# NeoR, a near-infrared absorbing rhodopsin

Matthias Broser [1✉], Anika Spreen[1], Patrick E. Konold[2], Enrico Schiewer [1], Suliman Adam [3], Veniamin Borin [3], Igor Schapiro [3], Reinhard Seifert[4], John T. M. Kennis [2], Yinth Andrea Bernal Sierra [1] & Peter Hegemann [1]

The *Rhizoclosmatium globosum* genome encodes three rhodopsin-guanylyl cyclases (RGCs), which are predicted to facilitate visual orientation of the fungal zoospores. Here, we show that RGC1 and RGC2 function as light-activated cyclases only upon heterodimerization with RGC3 (NeoR). RGC1/2 utilize conventional green or blue-light-sensitive rhodopsins ($\lambda_{max} =$ 550 and 480 nm, respectively), with short-lived signaling states, responsible for light-activation of the enzyme. The bistable NeoR is photoswitchable between a near-infrared-sensitive (NIR, $\lambda_{max} =$ 690 nm) highly fluorescent state ($Q_F = 0.2$) and a UV-sensitive non-fluorescent state, thereby modulating the activity by NIR pre-illumination. No other rhodopsin has been reported so far to be functional as a heterooligomer, or as having such a long wavelength absorption or high fluorescence yield. Site-specific mutagenesis and hybrid quantum mechanics/molecular mechanics simulations support the idea that the unusual photochemical properties result from the rigidity of the retinal chromophore and a unique counterion triad composed of two glutamic and one aspartic acids. These findings substantially expand our understanding of the natural potential and limitations of spectral tuning in rhodopsin photoreceptors.

[1] Institute for Biology, Experimental Biophysics, Humboldt-Universität zu Berlin, 10115 Berlin, Germany. [2] Department of Physics and Astronomy, Faculty of Science, Vrije Universiteit Amsterdam, De Boelelaan 1081, 1081 HV Amsterdam, The Netherlands. [3] Fritz Haber Center for Molecular Dynamics, Institute of Chemistry, The Hebrew University of Jerusalem, 9190401 Jerusalem, Israel. [4] Molecular Sensory Systems, Center of Advanced European Studies and Research (caesar), Ludwig-Erhard-Allee 2, 53175 Bonn, Germany. ✉email: matthias.broser@hu-berlin.de

Rhodopsins are widely distributed sensory photoreceptors that, among other functions, promote different levels of vision in prokaryotes, algae, and animals[1,2]. Although rhodopsins have been color tuned over a wide spectral range during evolution, as well as by molecular engineering[3], the limits of detection towards both ends of the spectrum into the ultra-violet (UV) and far-red ranges have not been systematically explored experimentally or by theoretical approaches[4].

In several flagellated fungal zoospores, photoorientation is mediated by homodimeric rhodopsin cyclases (termed RhGCs or simply RGCs; Fig. 1a), in which rhodopsins are directly linked to type III guanylyl cyclases for cyclic GMP (cGMP) upregulation upon illumination (cartoon in Fig. 1b)[5]. Here, we describe two RGCs (RGC1 and RGC2) of the fungus *Rhizoclosmatium globosum*[6], from the phylum Chytridiomycota (Fig. 1a and Supplementary Fig. 1), each containing a classic blue-green absorbing rhodopsin, which heterodimerize with another uncharacterized rhodopsin, RGC3 or neorhodopsin (NeoR), that we show has sensitivity in the near-infrared (NIR) spectrum.

## Results

### Functional characterization of *R. globosum* RGCs.

Phylogenetic analysis and multiple sequence alignment of the three *R. globosum* RGC rhodopsin modules with other microbial rhodopsins from distinct families revealed that RGC1/2 and RGC3(NeoR) from *R. globosum* separate as two distinct phylogenetic branches (Fig. 1a and Supplementary Fig. 1). In contrast the coiled-coil linker and the cyclase domain are highly conserved among all three proteins (Supplementary Fig. 2).

To characterize their function, we first expressed the three putative RGCs of *R. globosum* in *Xenopus laevis* oocytes, in conjunction with the cGMP-activated cyclic nucleotide-gated (CNG)-A2 ion channel from rat olfactory neurons[7]. Photocurrents were exclusively observed when RGC1 or RGC2 were co-expressed with NeoR and CNG-A2, while neither the combination of RGC1 with RGC2 nor individually expressed constructs showed any light response (Fig. 1c). This suggested that both RGC1 and RGC2 are only functional after forming heterodimers with NeoR (RGC1/NeoR and RGC2/NeoR; Fig. 1b). To our knowledge, heterodimerization has not been reported for any rhodopsin, although it does occur with the rhodopsin-related odorant receptors (ORs) in *Drosophila* olfactory sensory neurons[8]. Next, we expressed RGC1, RGC2, and NeoR individually and in different combinations together with an engineered cGMP-gated potassium-selective channel in mammalian hybrid cells (ND7/23). Once again, neither RGC1, RGC2, nor NeoR alone were able to produce any photocurrents when co-expressed with the channel in ND7/23 cells. In contrast,

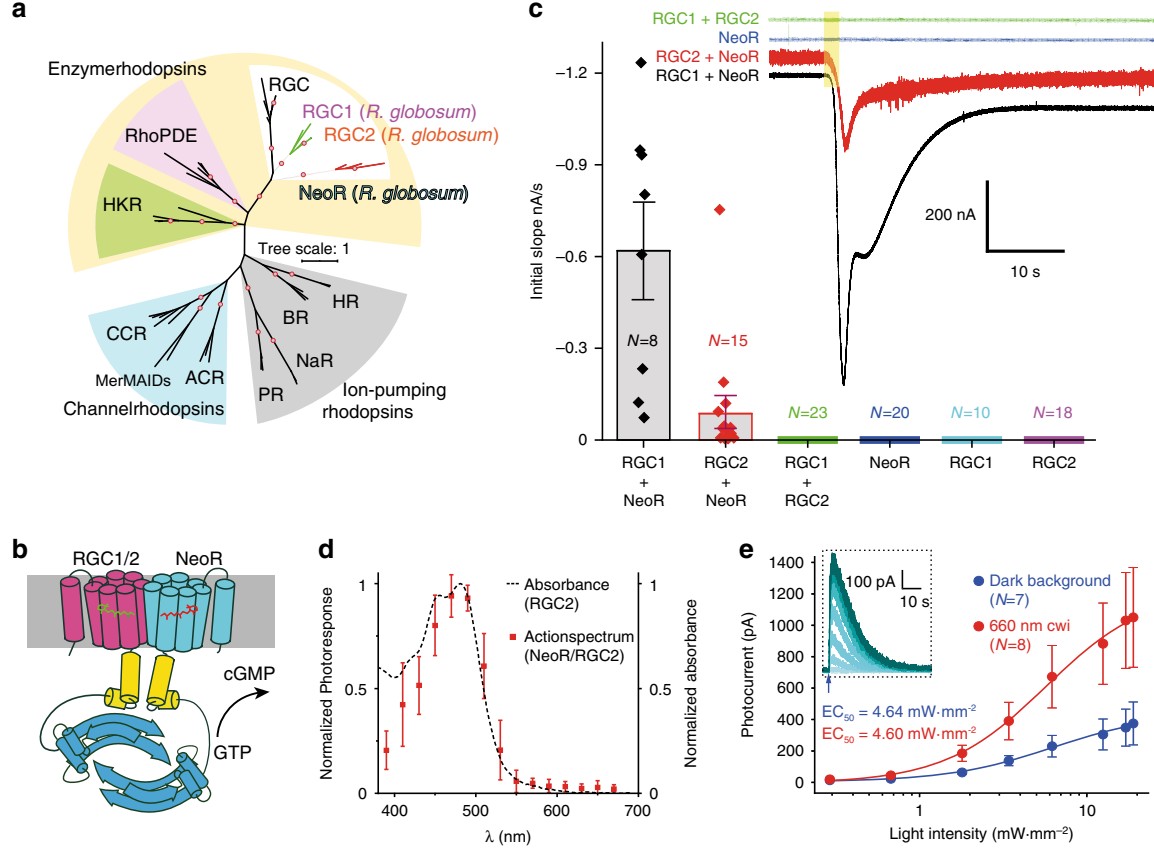

**Fig. 1 Phylogeny, model, and activity of the *Rhizoclosmatium globosum* rhodopsin-guanylyl cyclases (RGCs). a** Unrooted phylogenetic tree of microbial rhodopsins, with red circles indicating bootstrap values >90%; the scale bar represents average amino acid substitutions per site. **b** Model of *R. globosum* heterodimeric RGCs: rhodopsins, magenta (RGC1/2) and cyan (NeoR); cyclase, blue; and linker, yellow. **c** Representative currents from *Xenopus* oocytes expressing RGC1/2 and NeoR individually or in combination, in response to a 2 s light pulse (yellow box); initial photocurrent slopes are plotted as mean ± standard error (s.e.m.); *N* = number of biological replicates. **d** Photocurrent action spectrum (red squares; means ± standard deviation (s.d.), derived from three biological replicates) of RGC2/NeoR in ND7/23 cells, normalized to the maximum current underlayed by the RGC2 absorption. **e** Photocurrents (means ± s.e.m., with *N* = number of biological replicates) of RGC2/NeoR heterodimers generated by 10 ms blue-light (490 nm) flashes of different intensities (as visualized in the inset) in the absence (blue cycles) or presence (red cycles) of 660 nm background light (continuous wave illumination (cwi)).

RGC2/NeoR displayed photocurrents with maximal sensitivity in the blue, that were graded with the light-intensity with half-saturation EC50 = 4.6 mW mm$^{-2}$ (Fig. 1d, e). Although the action spectrum shows no photoresponse in the red spectral range (Fig. 1d), continuous background-illumination with 660 nm light resulted in significantly increased photocurrent amplitudes evoked by the application of 490 nm light flashes in a light titration experiment, while the light saturation behavior of the signal remained unaffected (Fig. 1e red dots, Supplementary Fig. 3a). Enlarged photocurrent amplitudes were also observed in single cell experiments upon pre-illumination with a 660 nm light pulse for 15 s (Supplementary Fig. 3b, c). RGC1/NeoR tended to precipitate in large intracellular clusters and were only poorly targeted to the plasma membrane (Supplementary Fig. 4a), with the consequence that no photoresponses were observed in ND7/23 cells.

**Spectral properties of the rhodopsin domains**. To better understand the spectral properties of the *R. globosum* RGCs, we expressed the rhodopsin fragments of all three proteins in human embryonic kidney cells (HEK-T) and insect cells (Sf21). The recombinant RGC1 (from HEK-T) and RGC2 (from Sf21) fragments were purified with low yield as red and yellow proteins with typical rhodopsin spectra showing maxima at 550 and 480 nm, respectively, the latter corresponding to the RGC2/NeoR-dimer action spectra measured in ND7/23 cells (Figs. 1e and 2a). NeoR expressed well in both systems and purified recombinant NeoR appeared turquoise or cyan (Inset Fig. 2a), with the main narrow band displaying an absorption maximum at 690 nm, a shoulder at 640 nm, and a second lower visible absorption around 400 nm. No other naturally occurring microbial rhodopsin ever showed a peak absorption beyond 610 nm[9], and the spectra from only a few animal rhodopsins have been predicted as such, including the goldfish and salamander red cones ($\lambda_{max}$ = 617 nm and 615 nm, respectively), both incorporating 3,4-dehydroretinal (A2-retinal) as chromophore[10]. Some of the many rhodopsins from crustacean shrimps, however, may have indeed maxima near 700 nm, a range that is far beyond human and any other

known animal vision[11]. However, these photoreceptor proteins remain uncharacterized, and their light-sensitive cofactors are unknown. Notably, the used *R. globosum* strain JEL800 was originally isolated from purified shrimp chitin, although the significance of this correlation is unclear[12].

**NeoR is a photochromic bistable rhodopsin with a highly fluorescent NIR-state**. We further found that the NeoR main absorption band disappears upon illumination with far-red light and gives rise to short-wavelength absorption, with a maximum at 367 nm (Fig. 2b). This UV-absorbing state is also thermally stable but can be completely reconverted into the NIR state by illumination with UV light (Fig. 2c, d), thereby disclosing that NeoR is a photochromic bistable rhodopsin, i.e., a rhodopsin with two thermally stable states that are spectrally distinct[2,13]. The spectral properties are similar for the NeoR full-length protein and the rhodopsin fragment in detergent and protein/lipid nanodiscs (Supplementary Fig. 5a–c). According to these results, the increased photocurrents observed for RGC2/NeoR hetero-dimers in ND7/23 cells by illumination with far-red light may be assigned to NeoR that is switched between the two states NeoR$_{690}$ and NeoR$_{367}$. However, the application of UV light (365 nm) required for reconversion to RGC2/NeoR$_{690}$ leads to strong enzyme activation and reduced patch quality, which hindered us from further investigating the reversibility of photocurrent modulation.

Both, the NIR absorption and the spectral band shape resemble phytochromes that use linear tetrapyrroles as the chromophore (Fig. 2e). However, in the absence of retinal no functional rhodopsin was produced when recombinant NeoR was expressed in insect cells, whereas supplementation of the growth media with 3,4-dehydroretinal (A2-retinal) resulted a NeoR with an absorption band even further bathochromic shifted to 759 nm (A2-NeoR; Fig. 2e) and well-preserved bimodal photoswitching. Such a large shift of 70 nm (1317 cm$^{-1}$) for A2 compared to A1 has never been observed for a rhodopsin to our knowledge[14,15]. Similar to other retinylidene proteins, denaturation of NeoR leads to the appearance of a 440 nm absorption band (Supplementary

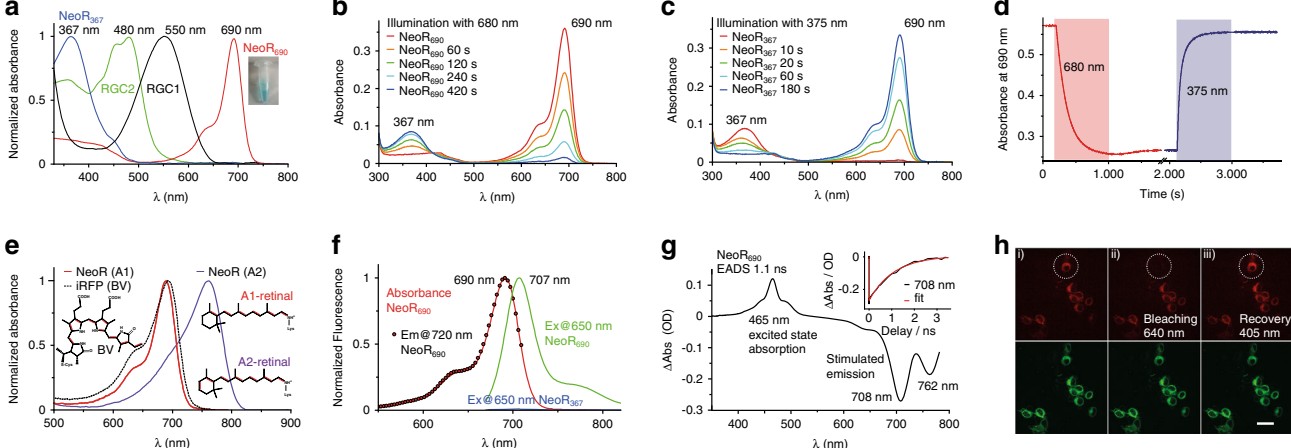

**Fig. 2 Spectral properties of purified rhodopsin domains from the *R. globosum* RGCs. a** Normalized absorption spectra of the purified recombinant rhodopsin domains from RGC1, RGC2, and NeoR (red and UV-form); picture of the purified NeoR sample, inset. **b** Reversible bleaching of NeoR$_{690}$ with 680 nm light. **c** Reversible bleaching of NeoR$_{367}$ with 375 nm light. **d** Kinetic traces of NeoR bleaching with 680 nm light and subsequent recovery by 375 nm illumination. **e** Absorption spectra of purified NeoR expressed in media supplemented with A1-retinal or 3,4-dihydro-retinal (A2), as compared to those from biliverdin (BV)-containing iRFP[50]. **f** Fluorescence spectra of NeoR$_{690}$: excitation (dots) overlaid with the absorbance (red) and emission (green). **g** Evolution-associated difference spectra (EADS) of fs–ns transient absorption spectra derived from pump-probe experiments, time trace at 708 nm, inset. **h** Representative confocal images of NeoR fluorescence (red) and the C-terminal eGFP tag (green) in ND7/23 cells. NeoR is reversibly bleached by illumination with 640- and 405 nm light. The reversible bleaching of NeoR fluorescence in single cells was repeated in three independent experiments using five fields of view each. Scale bar, 100 μm.

Fig. 6a–d), resembling the absorption of the protonated retinal Schiff base (RSBH$^+$) in water.

A further atypical property for a rhodopsin is the strong fluorescence of NeoR$_{690}$ with an emission maximum at 707 nm, a quantum yield of $\Phi_F = 0.2$ (20%) and a small Stokes shift of only 350 cm$^{-1}$ (17 nm, Fig. 2f and Supplementary Fig. 7). The small Stokes shift as well as the pronounced mirroring vibrational structure of the emission and absorption spectra indicate a close proximity between the Frank-Condon region and the minimum of the S$_1$ potential energy surface. This proximity implies that the chromophore is undergoing only a small geometrical or structural relaxation. Hence, we conclude that the presence of a barrier in the S$_1$ excited state prevents the photoisomerisation and promotes the excited-state lifetime to the nanosecond range[16]. We further investigated the excited-state dynamics of NeoR using pump-probe experiments and confirmed a long excited-state lifetime of $\tau = 1.1$ ns (Fig. 2g). In contrast, all other known rhodopsins have a negligible fluorescence, with $\Phi_F$ values in the order of 10$^{-5}$ for bovine rhodopsin and HKR1[17,18] up to $9 \times 10^{-4}$ for the Arch[19] and proteorhodopsin[20] proton pumps corresponding to excited-state lifetimes in the range of 100 fs to a few ps. Additionally, substantial bioengineering efforts to increase the fluorescence of microbial rhodopsins as voltage sensors or incorporation of retinal analogs have yielded maximal $\Phi_F$ values of 3.3%[21–23]. Here, we note that although we have not yet been able to observe a voltage dependency of NeoR fluorescence in mammalian cells, the NIR fluorescence can be utilized for monitoring RGC/NeoR expression in host cells, and this fluorescence may be switched *on* and *off* by alternating UVA and far-red light (Fig. 2h and Supplementary Fig. 4a, b). The great fluorescence brightness, which is the product of the large extinction coefficient, $\varepsilon_{690} = 129000$ M$^{-1}$ cm$^{-1}$ (Supplementary Fig. 6a), and high quantum yield $\Phi_F$ (0.2) suggests NeoR to be suitable for fluorescence-imaging in the NIR spectral range.

**The molecular basis of NeoR NIR absorption maximum.** The unique photochemical properties raise the question of how the protein environment in NeoR can shift absorption of the protonated retinyl imine (also named protonated retinal Schiff base,

RSBH$^+$) from 440 nm in solution to 690 nm, corresponding to an opsin shift of 250 nm (8235 cm$^{-1}$). Shortly after discovery of the light-driven proton pump bacteriorhodopsin as the first microbial rhodopsin (BR, $\lambda_{max} = 568$ nm), Warshel as well as Honig and coworkers proposed a point charge model to explain the 128 nm (5121 cm$^{-1}$) BR opsin shift, which was considered large at that time. This model involves two negative charges, one of which operates as a counterion near the RSBH$^+$ and the other one near the β-ionone ring[24–26]. Upon excitation an intramolecular charge transfer occurs between the Schiff base nitrogen and the β-ionone ring. Translocation of the charge has differential effects on the electronic ground- and excited-state energy levels, which provided a qualitative explanation for the origin of the opsin shift. While the proposed negatively charged residue close to the β-ionone ring was not found in any microbial rhodopsin, it is widely accepted that increasing the polarity near the ring still promotes opsin shifts[4,27,28].

In order to understand how NeoR enables NIR-light absorption, we generated a homology model of the protein. Since no structure with more than 17% sequence identity was available, we combined multiple crystal structures of various microbial rhodopsins matching different regions of NeoR (see SI for details). Despite the low-overall sequence similarity, many of the conserved residues that constitute the chromophore binding pocket in other microbial rhodopsins (e.g., BR) were also present in NeoR. Thus, we did not detect any obvious clues to immediately explain the NIR absorption of NeoR. Therefore, to further elucidate the molecular basis for the large NeoR opsin shift, we mutated the amino acid residues within a radius of ~5 Å from the retinal (Fig. 3a) by changing the polarity with only modest alteration of the structure and referring to analogous mutations carried out in BR (Supplementary Table 1). We found that mutation of residues along the polyene chain that are predicted to increase the polarizability of the NeoR retinal-binding pocket, namely H134Y, E141C/Q, and M192L, induce only small spectral shifts of <10 nm (Supplementary Table 1). More significant spectral shifts to the blue (up to ~30 nm) and reduction of fluorescence are achieved by mutation of S191A and T238P/A, which are located near the β-ionone ring (Fig. 3b–e). These effects are additive, with a 65 nm blue shift observed in the double mutant (S191A, T238A).

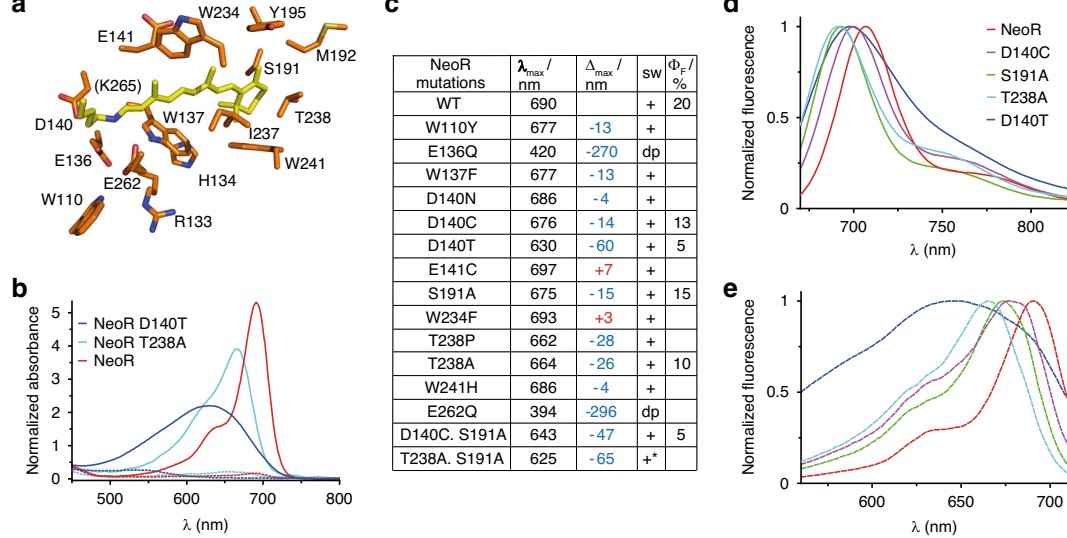

**Fig. 3 Molecular model and spectral features of NeoR mutants. a** Homology model of the NeoR retinal-binding pocket, with the amino acids targeted by mutagenesis drawn as sticks. **b** Absorption spectra of NeoR mutants; spectra are normalized to the respective UV-form. **c** Spectral features of NeoR mutants; sw, bimodal switchable; dp, deprotonated chromophore; $\Phi_F$, fluorescence quantum yield (in %). **d** Emission spectra (at 650 nm), and **e** excitation spectra (at 720 nm) of wild-type (WT) NeoR and NeoR mutants.

| NeoR mutations | $\lambda_{max}$/ nm | $\Delta_{max}$/ nm | sw | $\Phi_F$/ % |
|---|---|---|---|---|
| WT | 690 | | + | 20 |
| W110Y | 677 | -13 | + | |
| E136Q | 420 | -270 | dp | |
| W137F | 677 | -13 | + | |
| D140N | 686 | -4 | + | |
| D140C | 676 | -14 | + | 13 |
| D140T | 630 | -60 | + | 5 |
| E141C | 697 | +7 | + | |
| S191A | 675 | -15 | + | 15 |
| W234F | 693 | +3 | + | |
| T238P | 662 | -28 | + | |
| T238A | 664 | -26 | + | 10 |
| W241H | 686 | -4 | + | |
| E262Q | 394 | -296 | dp | |
| D140C. S191A | 643 | -47 | + | 5 |
| T238A. S191A | 625 | -65 | +* | |

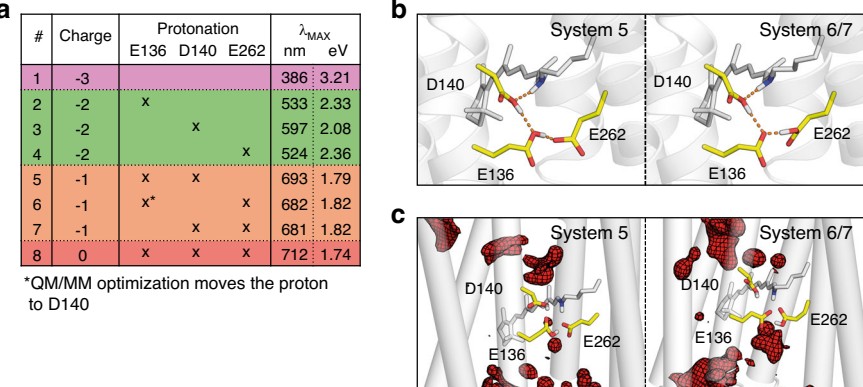

**Fig. 4 QM/MM Modeling of NeoR. a** Overview of QM/MM excitation energy calculations of eight systems (#1–8) with varying protonation of E136, D140 and E262 as indicated. The total charge of the three carboxylates in each system is given. $\lambda_{MAX}$ refers to the absorption maxima as derived from excited-state energy calculation. **b** Active site from QM/MM-optimized models, with hydrogen bonds represented as dashed lines. **c** Water intrusion into NeoR as observed during subsequent 300 ns MD simulations with water densities visualized as red meshes. The retinal and amino acid sidechains are taken from the initial structure of the MD simulations.

Nevertheless, the spectral changes are in the range of analogous mutations in many other microbial rhodopsins or even smaller and do not explain the NIR absorption of NeoR[3,22,27–32].

We found the active site, defined by the RSBH[+] and its direct protein- and possibly water-environment to be by far the most critical region for the far-red shifted absorption in NeoR. The two RSBH[+] aspartate counterions, D85 and D212, in BR are replaced by glutamates, E136 and E262, in NeoR. Mutation of either one of these to Gln results in chromophore deprotonation (Fig. 3c), whereas Asp substitution completely prevents retinal binding in both cases (Supplementary Table 1). In contrast, when the unique third Asp, D140, which is predicted to interact with E136, is replaced by Thr (T89 in BR), the spectrum becomes broad and strongly blue shifted by 60 nm (Fig. 3b), while the photoreceptor retains its bistability.

The D140 homolog position occupied by Thr in many microbial rhodopsins has been widely disregarded by experimentalists but was suggested by theoreticians to contribute to the proton transfer from RSBH[+] to the acceptor Asp or Glu in rhodopsin pumps and channels[33–35]. Moreover, replacement of this Thr in the proton pump Arch to Cys, T99C, in combination with the D95E mutation of the primary proton acceptor shifted the absorption from 556 nm to 626 nm thus creating the largest bathochromic color shift and the most red-shifted microbial rhodopsin which, however, remained unexplained[21]. Previous studies on the Chrimson channelrhodopsin have also demonstrated the importance of residues at the D140 homologous position by inducing color shifts beyond 600 nm upon modification of this location[31]. The most extended color tuning study on a retinal protein was carried on the human cellular retinol binding protein II (hCRBPII) that was converted into a retinal binder[36]. Mutagenesis of the retinal-binding pocket lead to a protein with absorption at 644 nm[37,38], a strong bathochromic shift that depends on the even distribution of the electrostatic potential across the entire polyene. However, one has to point out that hCRBPII is missing any RSBH[+] counterion and our data demonstrate that the situation is clearly different in NeoR with low importance of the binding pocket and a main contribution of the active site.

### Hybrid quantum mechanics/molecular mechanics simulations of NeoR.
The contribution of the critical E136, D140, and E262 to the spectral properties of NeoR was further investigated by a hybrid quantum mechanics/molecular mechanics (QM/MM) approach, with the retinal and all three sidechains included in the QM region. The treatment of the carboxylates by a quantum chemical method allowed to include polarization effects which is an improvement compared to the static point charges typically chosen in the MM region. Further, it provides a flexibility with respect to the protonation state because a proton can be transferred among the titratable residues in the QM region. Since the protonation states of these three titratable residues could be decisive for the spectral tuning, we systematically studied four scenarios where the three potential counterions have a total charge of 0, −1, −2, and −3 (Fig. 4a). Irrespective of the protonation patterns, decreasing the number of charged carboxylates systematically red-shifted the spectrum (Fig. 4a). Notably, the excited-state energy decreased substantially upon reducing the number of negatively charged carboxylates from 2 to 1, while only a small further red shift was observed for the protonation of all three carboxylates (Fig. 4a). The latter, fully protonated configuration, however, turned out to be unstable during molecular dynamics (MD) simulations of the entire protein embedded in the membrane. In contrast, the models derived from QM/MM with only one carboxylate, either E136 or E262 deprotonated (systems 5 and 6/7, Fig. 4) were stable during long MD runs of 300 ns. Hence, the strong red shift obtained from QM/MM simulations as well as the stability during a classical MD simulation points towards a single deprotonated glutamate.

Analysis of the QM/MM optimized geometries shows that the negative charge is located relatively far from the RSBH[+] (Fig. 4b). Hence, the ground state energy is less stabilized with respect to the electronically excited state, in accordance with the point charge model.

All active-site models (system 1–8) have been subjected to extended MD simulations with the protein embedded in the membrane. Interestingly, low water density was found close to the RSBH[+] for models with one deprotonated counterion (system 5–7, Fig. 4c), but significant water content was present in models of higher total charge (system 1–4). As water promotes extensive ground state dynamics in other microbial rhodopsins[39], its absence would explain the narrow spectrum observed for NeoR and thus support an active site model with only one charged carboxylate.

### Discussion
Our data indicate that the specific spatial organization of the RSBH[+] environment, which enables an unusual delocalization of a single negative charge, is responsible for the narrow red-shifted

absorption. We note, however, that structural data will be needed to obtain a more detailed understanding of the chromophore-protein interaction. The polarizability of the retinal-binding pocket determines most of the color shift in the retinal-binding protein hCRBPII[40], but only fine tunes the spectral properties of the chromophore in NeoR, similarly as reported from QM/MM calculations on animal rhodopsins[41]. We predict that similarly rigid chromophore constellations will be found in other enzyme or sensory rhodopsins, but not in ion-transporting rhodopsins—pumps or channels—for which access of water to the active site is essential for function. Indeed, we found a similar active site constellation in an RGC from the related fungus *Chytriomyces confervae*[42], which shares sequence features with the *R. globosum* NeoR. This *Cc*NeoR1 also exhibits a bistable photochromic chromophore with an absorption maxima at 677 nm and 362 nm. It forms functional heterodimeric photorecpytors with *Cc*RGC1 from the same organism (Supplementary Fig. 9)

Admittedly, although we observed a promotion of photo-currents upon pre-illumination with NIR-light, the mechanism of how NeoR is modifying the photoreceptor response remains unclear. We note that the $NeoR_{690}$ absorption overlaps strongly with the expected broad fluorescence spectrum of RGC1 and RGC2 (Supplementary Fig. 10) thereby potentially allowing Förster resonant energy transfer (FRET) from RGC1/2 to $NeoR_{690}$ within the RGC heterodimer. Such energy transfer would reduce the photoresponse in the $NeoR_{690}$ state, but would disappear after NIR illumination and conversion of $NeoR_{690}$ to $NeoR_{367}$. A FRET calculation (as detailed in the SI) indicated energy transfer with expected lifetimes in the order of 1–10 ps competing with the sub-ps to ps isomerization of RGC1/2. As an alternative we could imagine an allosteric down-modulation of the activity by $NeoR_{690}$, which is relieved by conversion to $NeoR_{367}$. It is known from other heterodimeric type III cyclases that enzyme activation can follow a complex mechanism influenced by multiple stimuli, as the synergistic activation of G-protein and $Ca^{2+}$/calmodulin-binding described for mammalian pseudo-heterodimeric membrane-bound adenylyl cyclase[43]. Chytridiomycota fungi are ubiquitous microbes present in various aquatic habitats and show a high diversity of saprobiotic and parasitic lifestyles[44]. Thus, a more complex photo-perceptive system might be requested, which includes sensing of far-red over UVA, whereas direct light-activation of the cyclase can be switched between green (RGC1) and blue-light (RGC2) by differential expression. The three proteins—RGC1, RGC2, and NeoR—together form a photosensory system that covers the entire visible spectrum, expanding into both directions, namely UVA and NIR (Fig. 2a) and thereby blurring the spectral boundaries between rhodopsins and phytochromes.

During the last fifteen years microbial rhodopsins have attained much attention due to their employment in optogenetic experiments especially in the neurosciences for the modulation cellular activities with light. Especially Channelrhodopsins are widely used for neuronal stimulation with unprecedented spatial and temporal precision. Despite the enormous impact of this opto-genetic approach, one of the major limitations of this technique is the low penetration depth of visual light in biological tissues. In contrast to all known rhodopsins, both absorption and fluorescence of NeoR are within the NIR window (Supplementary Fig. 11), marking this photoreceptor an excellent starting point for developing optogenetic tools that are excited by NIR-light with as-yet-unreached penetration depths, which will enhance our capability for non-invasive in vivo experiments.

## Methods

**Protein expression and purification.** For insect cell expression (Sf21 cells (Gibco, Thermo Fisher Scientific, Waltham, MA, USA)), the rhodopsin domains of NeoR (UniProtKB A0A1Y2CSJ0; aa: 1–279) and rhodopsin-guanylyl-cyclase (RGC)1 and 2 (UniProtKB A0A1Y2CSF8; aa: 1-320 and UniProtKB A0A1Y2CSL9; 1-319, respectively) from *Rhizoclosmatium globosum* were cloned into the pFastBac1 vector (Invitrogen, Thermo Fisher Scientific, Waltham, MA, USA), with a C-terminal strepII-tag for NeoR and His(8)-tags for RGC1/2. Further NeoR full-length sequence (UniProtKB A0A1Y2CSJ0; aa: 1-509) was cloned into the pFast-Bac1 vector with C-terminal His(8)-tag. Bacmid preparation, transfection, virus production, and expression were performed following the instruction of the manufacturer (Invitrogen)[45]. Crude membranes were prepared according to a protocol developed for G-protein-coupled receptors (GPCRs)[46]. Membranes were solubilized at 4 °C for at least 4 h in binding buffer, containing 50 mM HEPES (pH 7.4), 200 mM NaCl, 1 mM phenylmethylsulfonyl fluoride (PMSF), and 5 μM all-*trans* retinal, with 1.5% n-dodecyl β-D-maltoside (DDM)/0.3% cholesteryl hemi-succinate (CHS), followed by centrifugation at 30,000 x g for 15 min at 4 °C. Full-length NeoR protein was solubilized overnight at 4 °C in a mixture of DDM/CHS/ 1,2-Dimirystoyl-sn-glycerol-3-phosphocholine(DMPC)/N,N-dimethyl-n-dodecy-lamine N-oxide (LDAO) in a final concentration of 2%/0.5%/0.01%/0.25% (w/v)[45], but otherwise handled similar as RGC1/2. For His-tagged proteins, solubilization buffer was supplemented with 30 mM imidazole, and the proteins were purified using 5 ml HisTrap columns (GE Healthcare, Chicago, IL, USA). Supernatants were applied to the columns, washed with washing buffer (50 mM HEPES (pH 7.4), 200 mM NaCl, 0.02% DDM/0.004% CHS) supplemented with 50 mM Imidazole, eluted with washing buffer with 500 mM imidazole, and concentrated in washing buffer. The supernatant of the solubilized NeoR-rhodopsin fragment was applied to a 5 ml Strep-Tactin®XT Superflow® column (IBA GmbH, Göttingen, Germany) and washed with ten column volumes of washing buffer. Protein was eluted with 2X BTX-buffer, containing 100 mM biotin (IBA GmbH), supplemented with 0.02% DDM/0.004% CHS, and colored fractions were pooled and concentrated in washing buffer (Amicon Ultra Centrifugal Filter, molecular weight cutoff (MWCO) 100 kDa, Merck Millipore, Burlington, MA, USA).

**Protein purification in SMALPS.** Styrene-maleic acid copolymer (SMA 3:1; 20% (w/w) Xiran sl25010 s25, Polyscope Polymers B.V., 6161 CZ Geleen,The Nether-lands) was extensivly dialyzed against 50 mM HEPES pH 7.4, 150 mM NaCl. Crude membranes of NeoR-rhodopsin fragment were solubilized at room temperature for 16 h using 2% SMA (w/w) in HEPES buffer with 1 mM PMSF. After ultra-zentrifugation at 146,550 x g for 30 min at 4 °C using a Beckman 70.1 Ti rotor, the supernatend was applied to a Strep-Tactin®XT 4Flow® gravity flow column (IBA GmbH, Göttingen, Germany), washed with 50 mM HEPES pH 7.4, 150 mM NaCl and eluted with 2xBTX buffer (IBA GmbH). The formation of SMALPS was proofed by dynamic light scattering (DynaPro NanoStar, Wyatt Technology Cor-poration, Santa Barbara, CA 93117, USA) confirming the expected particle size of ~11 nm.

**Mutant screening in human embryonic kidney (HEK)-T cells.** For expression in mammalian cells, the rhodopsin domains of NeoR (UniProtKB A0A1Y2CSJ0; aa: 1-279) and RGC1/2 (UniProtKB A0A1Y2CSF8; aa: 1-320/UniProtKB A0A1Y2CSL9; aa: 1-319, respectively), as well as the full-length proteins from *Chytriomyces confervae* (NeoR1: UniProt A0A507FHL0 and RGC1: UniProt A0A507F303) were cloned into the EGFP-C1 vector (Clonetech, Takara Bio, Kusatsu, Shiga Prefecture, Japan) with C-terminal 1D4-tags (TETSQVAPA). Site-directed mutagenesis was performed according to the QuikChange protocol from Agilent, using *Pfu* polymerase (Agilent Technologies, Santa Clara, CA, USA) using the primers listed in the Supplementary Data 2. For each construct, two petri dishes (10-cm diameter) of HEK-T (ECACC 12022001, grown in Dulbecco's Modified Eagle's Medium supplemented with 1% fetal bovine serum and Penicillin–Streptomycin (Gibco, Thermo Fisher Scientific)) cells were transfected using TurboFect (Thermo Fisher Scientific), according to the manufacturer's protocol. At 2 days post transfection, cells reached >90 % confluence and were harvested and washed with Dulbecco's phosphate-buffered saline (DPBS; Gibco, Thermo Fisher Scientific), containing 2X complete protease inhibitor (Merck, Darmstadt, Germany). The cell pellets were resuspended in 1-ml binding buffer (see above), and membrane proteins were solubilized with 1.5% DDM/0.3% CHS for 4 h at 4 °C. Insoluble material and cell debris were separated by centrifugation at 21,000 x g for 20 min at 4 °C; 150 μl 1D4 affinity beads (CUBE Biotech GmbH, Monheim, Germany) were added to the supernatant, and this was incubated at 4 °C overnight. Beads were washed three times with 1-ml washing buffer (see above), and bound proteins were eluted in two steps with washing buffer, supplemented first with 0.3- and then with 1 mg/ml 1D4 peptide.

**TEVC measurements in Xenopus oocytes.** For two-electrode voltage clamp (TEVC) experiments, cRNAs coding for NeoR and RGC1/2 were synthesized from linearized DNA (*Nhe*I), using the mMESSAGE mMACHINE TM T7 Transcription Kit, according to the manufacturer instructions (Invitrogen). Between 2 and 15 ng of total cRNA from either a single construct or a mixture of two were then injected into *Xenopus laevis* oocytes, together with 5 ng cRNA coding for the cGMP-sensitive cyclic nucleotide-gated (CNG)A2 ion channel from rat olfactory neurons (gb: 6978671, NP_037060.1; $K_{1/2}^{cAMP} = 36$ μM, $K_{1/2}^{cGMP} = 1.3$ μM)[7]. Oocytes were incubated at 18 °C in Ringer's solution, containing 96 mM NaCl, 5 mM KCl,

0.1 mM $CaCl_2$, 1 mM $MgCl_2$, and 5 mM HEPES (pH 7.5), supplemented with 1 μM all-*trans* retinal, for 3–5 days. TEVC measurements at −40 mV were performed using a TURBO TEC-03X amplifier (NPI Electronic GmbH, Tamm, Germany), pCLAMP v. 9.0 software (Molecular Devices, San Jose, CA, USA), a XBO 75 W Xenon lamp (Osram, Munich, Germany), and a UNIBLITZ LS3 shutter (Vincent Associates, Rochester, NY, USA). The initial slope of the photocurrents was linearly fitted in Clampfit v. 10.7 (Molecular Devices). A similar experimental approach was used to investigate the functioning of *Cc*NeoR1 and *Cc*RGC1 from *Chytriomyces confervae*. The protocols for animal maintenance and oocyte harvesting were approved by the Federation of European Laboratory Animal Science Associations (Berlin, Germany).

**UV–Vis absorption and fluorescence spectroscopy of purified protein**. Absorption spectra of purified proteins (expressed either in insect cells or in HEK-T cells) were obtained in buffer containing 50 mM HEPES (pH 7.4), 200 mM NaCl, and 0.02% DDM/0.004% CHS, with 2% (w/v) (final concentration f.c.) sodium dodecyl sulfate (SDS) or 100 mM HCl (f.c.) added for protein denaturation, using a Shimadzu UV-2000 photospectrometer with UVProbe v2.34 (Shimadzu Corporation, Kyoto, Japan). Transient UV–Vis absorption measurements were obtained with a femtosecond pump–probe setup utilizing a Ti:sapphire amplified laser system[47,48]. A $CaF_2$ plate on a homemade moving stage was used for supercontinuum white light generation, which was focused and overlapped with the pump beam in the sample. The probe beam was then spectrally dispersed and detected in a multichannel detection system (Entwicklungsbüro Stresing, Berlin, Germany), comprises a prism spectrograph and a 1024-pixel back-thinned full-frame transfer (FFT)–charge-coupled device (CCD) detector (S7030-1006, Hamamatsu Photonics, Hamamatsu City, Shizuoka Prefecture, Japan). The time delay was varied up to 3.5 ns, and the central wavelength and power of the pump beam were set at 650 nm and 250 nJ, respectively. The instrument response function was ~100 fs. Global analysis fitting was performed for the transient absorption spectra using the Glotaran 1.5.1. program[49]. Fluorescence spectra were recorded on a Horiba FluoroMax 4 spectrometer with FluorEssence™ 2.5.2 (HORIBA Instruments Inc., NJ, USA). To determine fluorescence quantum yields (QY), the integrated fluorescence intensity of the sample was plotted over the absorbance at the excitation wavelength (650 nm); iRFP, QY 5.9%[50] and Cy 5.5, QY 20% (Molecular Probes, Eugene, OR, USA) were used as references.

**Confocal microscopy**. The full-length NeoR gene sequence with a C-terminal eGFP tag was transfected into ND7/23 (ECACC 92090903) cells using FuGENE HD Transfection Reagent (Promega, Madison, WI, USA), according to the manufacture instructions. For coexpression of NeoR and RGC1/2 in ND7/23-cells full-length NeoR was cotransfected together with either full-length RGC1 or RGC2 C-terminally tagged with YFP. Images were recorded by spinning disc confocal microscopy, using an Olympus IX83 Inverted Microscope (Shinjuku, Tokyo, Japan) with a Yokogawa CSU-W1 Confocal Scanner Unit (Musashino, Tokyo, Japan) and an EMCCD Camera (Andor iXon Ultra 888) with a 60X/1.2 NA water immersion objective. Four-hundred eighty-eight nanometer laser light was used for the detection of GFP and YFP fluorescence, while intrinsic fluorescence of NeoR was excited with 640 nm laser light, and fluorescence recovery was achieved by 405 nm laser illumination.

**Characterization of the engineered cGMP-activated ion channel reporter**. SthK is a potassium-selective cyclic nucleotide-gated channel from *Spirochaeta thermophila* with a high preference for cAMP over cGMP. Modification of the C-linker and the cyclic nucleotide binding domain allow us to increase the ability of cGMP to open the channel. When expressed heterologously in Chinese Hamster Ovary (CHO) cells the generated channel mutant produces large whole-cell currents in the presence of 10 μM cGMP (Supplementary Fig. 12). CHO cells were transfected with using Lipofectamine 2000 (Thermo Fisher Scientific). Recordings were performed 1–2 days post transfection. Cells were bathed in a solution containing (in mM) 140 NaCl, 5.4 KCl, 1.8 $CaCl_2$, 1 $MgCl_2$, 10 Glucose, 5 HEPES, pH 7.4 (NaOH). The pipette solution contained (in mM) 130 KAsp, 10 NaCl, 2 $MgCl_2$, 1 EGTA, 2 $Na_2ATP$, 10 HEPES, pH 7,4 (KOH).

**Whole-cell voltage clamp recordings in ND7/23 cells**. Patch pipettes with a resistance of 2–3 MΩ were prepared from borosilicate capillaries (GB150P-9P; Science Products GmbH, Hofheim am Taunus, Germany), using the P1000 Micropipette Puller (Sutter Instrument Co., Novato, CA, USA). ND7/23 cells were transfected with RGC2/NeoR and the mutated cGMP-gated potassium channel described above using FuGENE HD Transfection Reagent (Promega). Recordings were made at room temperature in the whole-cell configuration, using an Axon MultiClamp 700B Amplifier. The signals were digitized with the Axon Digidata 1550 A Low-Noise Data Acquisition system (Molecular Devices) and recorded with pCLAMP v10.4 software. The whole-cell recordings had a minimum membrane resistance of 500 MΩ (usual > 1 GΩ) and an access resistance below 10 MΩ. Data were acquired at 5 kHz and filtered at 2 kHz. The intracellular solution contained the following (in mM): 17.8 HEPES, 135 KGluc, 4.6 $MgCl_2$, 4 MgATP, 0.3 NaGTP, 1 EGTA, 12 $Na_2$phosphocreatine, and 50 phosphocreatine kinase, pH 7.3.

The extracellular solution contained (in mM):10 HEPES, 140 NaCl, 2.4 KCl, 2 $CaCl_2$, 4 $MgCl_2$, and 10 glucose, pH 7.4. Action spectra were measured on an Axiovert 100 Carl Zeiss microscope (Oberkochen, Germany). Light was delivered through an Objective W Plan-Apochromat 40X/1.0 DIC objective (Carl Zeiss), with a Polychrome V light source (TILL Photonics, Hillsboro, OR, USA); the illumination field was 0.066 $mm^2$. The half bandwidth was set to ±7 nm for all data points, and light exposure time was controlled with a shutter system (VS25 and VCM-D1, Vincent Associates). In order to obtain the same photon irradiance for all wavelengths (photon count = $2.34 \times 10^{13}$ photons $s^{-1}$), a motorized neutral density filter (NDF) wheel (Newport Corporation, Irvine, CA, USA) was placed between the Polychrome V and the microscope. For light titration the CoolLED pE-4000 illumination system (CoolLED, Andover, UK) was coupled to an Olympus IX73 Inverted Microscope for light stimulation through the 60X objective. In light titration experiments a 490-nm light-emitting diode (LED) was used for RGC2/NeoR activation by 10 ms light flashes. Light-intensity dependent activation curve was made using 0.3, 0.7, 1.8, 3.4, 6.2, 12.5, 17.1 and 19.0 mW $mm^{-2}$ in presence or absence of 660 nm LED background light provided by the CoolLED pE-4000. Photocurrents (3 sweeps, 10 ms, 490 nm, 12.5 mW $mm^{-2}$ activation light) were further measured before and after 15 s illumination with 660 nm light within the same cell using this setup. 660 nm illumination for 15 s was sufficient for complete conversion of $NeoR_{690}$ to $NeoR_{367}$ as determined by the decrease of NeoR fluorescence. For seven independent experiments the averaged peak photocurrents of each cell, before and after red light treatment, as plotted in Supplementary Fig. 3b were subjected to a one-sided, paired student's $t$-test implemented in Origin 2017 (Originlab, Northampton, MA) resulting in $t = -3.6458$, d.f. = 6, $p = 0.00538$.

**Phylogenetic analysis**. The amino acid sequences of various microbial rhodopsins representing distinct families were obtained, including cation-conducting channelrhodopsin (CCR), metagenomically discovered anion-conducting channelrhodopsin (MerMAIDs)[51], proteorhodopsin (PR), sodium-pumping rhodopsin (NaR), bacteriorhodopsin (BR), halorhodopsin (HR), histidine-kinase rhodopsin (HKR), rhodopsin phosphodiesterase (RhoPDE), and rhodopsin guanylyl-cyclase (RGC) (for sequence information see Supplementary Data 1). These were aligned using ClustalO (1.2.4_1) and trimmed with TrimAl (1.4.1), and the phylogenetic tree was constructed with PhyML (3.1_1) in NGPhylogeny.fr (https://ngphylogeny.fr)[52] and visualized by iTOL (5.6.3)[53].

**Homology modeling**. Classical molecular dynamics (MD) simulations and computations of excitation energies were based on a homology model of NeoR (rhodopsin residues: A106–S287). The model was generated using the MODELLER (version 9.20) program (https://salilab.org)[54]. Multiple retinal proteins have served as a basis for the model generation: *Haloquadratum walsbyi* bacteriorhodopsin (*Hw*BR, PDB ID: 4QI1)[55], *Halobacterium salinarum* bacteriorhodopsin (*Hs*BR, PDB ID: 5ZIM)[56], Archærhodopsin-3 (AR3, PDB ID: 6GUX), Sensory Rhodopsin II (SRII, PDB ID: 5JJE)[57], *Acetabularia* Rhodopsin I (ARI, PDB ID: 5AX0)[58], Archærhodopsin-2 (AR2, PDB ID: 3WQJ)[59], Blue-light-absorbing proteorhodopsin (BPR, PDB ID: 4JQ6)[60], *Exiguobacterium sibiricum* rhodopsin (ESR, PDB ID: 4HYJ)[60], Channelrhodopsin Chrimson, PDB ID: 5ZIH)[31], *Chlamydomonas reinhardtii* channelrhodopsin-2 (*Cr*ChR2, PDB ID: 6EID)[61], Channelrhodopsin-1/−2 chimæra (C1C2, PDB ID: 3UG9)[62], *Krokinobacter eikastus* rhodopsin-2 (KR2, PDB ID: 6RF6)[63], *Gloeobacter* rhodopsin (GR, PDB ID: 6NWD)[64] and *Coccomyxa subellipsoidea* rhodopsin (CsR, PDB ID: 6GYH)[65]. Their sequences were aligned with ClustalO[66]. Standard protonation states were chosen for all amino acid residues of the model, except for the counterions E136, D140, and E262, for which all eight possible permutations of deprotonation and protonation were examined. Eight inner-protein waters were added based on hydration analysis using the DOWSER 0.26 plugin[67] in VMD 1.9.3 (https://www.ks.uiuc.edu)[68].

**Computation of excitation energies**. The eight protonation states of the NeoR homology model were minimized with a hybrid quantum mechanics/molecular mechanics (QM/MM) approach in the ChemShell 3.7.0 software package (https://www.chemshell.org)[69,70]. The MM region was described by the CHARMM36 protein force field[71,72] and the TIP3P water model[73]. To examine all possible starting locations of the counterion protons, i.e., on either of the carboxylate oxygens and in *cis/trans* conformation, we used a smaller QM region consisting only of the retinal chromophore and the side chain of K266 (see Supplementary Methods for details). The QM region of the final models consisted of the retinal chromophore and the sidechains of E136, D140, E262, and K266. The QM part of the minimization was done at the B3LYP/cc-pVDZ level of theory[74–76]. Excitation energies were computed in the framework of QM/MM using the algebraic diagrammatic construction scheme to second order with the resolution-of-identity (RI) approximation (RI-ADC(2))[77–80] and cc-pVDZ with the default auxiliary basis set for RI[81]. The Turbomole (version 7.3) program was used for the RI-ADC (2) method (http://www.turbomole.com).

**Molecular dynamics simulations**. The NeoR model was embedded in a hydrated 1-palmitoyl-2-oleoyl-sn-glycero-3-phosphocholine (POPC) lipid membrane using CHARMM-GUI[82–85]. The complete system had dimensions of 90 × 90 × 104 $Å^3$

and consisted of ~78,000 atoms (~15,000 water molecules and 211 lipids). NaCl was added to neutralize the system at a concentration of 0.15 M.

Two simulations were performed for each possible protonation state of E136, D140, and E262, resulting in a total of 16 MD trajectories. The MD simulations were run with the AMBER16 (https://ambermd.org)[86] software package employing a Langevin dynamics scheme, and the CHARMM36 protein and lipid force field[71,72,87], the TIP3P water model[73] and the ion parameters by Roux and coworkers[88]. Bonds involving hydrogens were constrained with SHAKE[89]. The system was heated from 100 K to 300 K and equilibrated under constant volume (NVT) during the first 750 ps of equilibration. The remainder of the equilibration (29.25 ns) and the production run were performed under constant pressure (NPT) of 1 bar. Initially, the heavy atoms had been harmonically restrained. These restraints were gradually released during the equilibration. After the equilibration, the integration step was changed from 1 fs to 2 fs, and a production run of 300 ns was performed. Coordinate snapshots were saved every 10 ps.

**Reporting summary**. Further information on research design is available in the Nature Research Reporting Summary linked to this article.

## Data availability
Data supporting the findings of this manuscript are available from the corresponding author upon reasonable request. Selected plasmids are available from AddGene. Source data are provided with this paper.

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

## Acknowledgements

We thank our technicians, Melanie Meiworm, Maila Reh, and Sandra Augustin, for their technical assistance. We also thank Oded Béjà, Peter Hildebrandt, and Ulrike Scheib for helpful discussion, Thomas Korte for microscopy support, and Wolfgang Bönigk for cloning support. Further we gratefully acknowledge the contribution of Saumik Sen and Rajiv K. Kar to the simulations. This work was supported by the German Research Foundation (DFG, SFB1078 (Grant No. 221545957) and Leibniz Project, PH) and the European Research Council (ERC) (Grant No. 693742 "MERA" and Grant No. 767092 "Stardust" (PH)). P.H. is a Hertie Professor for Neuroscience and supported by the Hertie Foundation. A.S., Y.A.B.S., and R.S. were supported by the German Research Foundation (DFG, SPP 1926 (Grant No. 315193289). S.A. thanks the Minerva Foundation for a postdoctoral fellowship. I.S. thanks the SFB1078 for support within the Mercator program and gratefully acknowledges funding by the European Research Council (ERC) under the European Union's Horizon 2020 research and innovation program (Grant No. 678169 "PhotoMutant").

## Author contributions

M.B. discovered NeoR and with P.H. conceived the project and designed the experiments. M.B. conducted most of the experiments, A.S. and Y.A.B.S. performed electro-physiological experiments in ND7/23 cells, R.S. engineered and characterized the mutated reporter-channel, S.A. constructed the homology model, performed MD simulations, QM/MM optimization and excitation energy calculations, E.S. constructed an initial structural template for the mutation study and contributed to the TEVC measurements, and P.E.K. and J.T.M.K. conducted time-resolved pump-probe spectroscopy and performed FRET calculations. S.A., V.B., and I.S. have analyzed and interpreted the results of the simulations. Technicians: M.M. expressed protein in insect cells, M.R. measured photocurrents in *Xenopus* oocytes, S.A. expressed and purified proteins from HEK cells, M.B. and P.H. wrote the manuscript, with contributions from all authors.

## Funding

## Competing interests

The authors declare no competing interests.
