## [Peer Review File · Nature Communications]

REVIEWER COMMENTS

Reviewer #1 (Remarks to the Author):

The manuscript reports a microbial rhodopsin sensitive to light in the near-infrared (NIR) region called NeoR. The study showed that NeoR forms heteromeric dimers with rhodopsin cyclases, RGC1 and RGC2, respectively, and activates the functions of light-sensitive cyclases in the heterodimeric forms. RGC1 and RGC2 exhibit photo-absorptions centered at 550 nm and 480 nm, respectively. The peak of the absorption spectrum of RGC2 resembles that of the action spectrum, indicating that RGS2 is responsible for the photo-detection for the cyclase function. In contrast, NeoR forms photochromic bistable states with absorption maxima at 690 nm and 367 nm, respectively, and the NIR state exhibits a strong fluorescence. Molecular mechanism of the NIR absorption of NeoR was investigated through homology modeling, mutagenesis experiments and theoretical calculations with a hybrid QM/MM method and MD simulations. The study suggested that Asp140 in the vicinity of the protonated Schiff base of the chromophore and one of the counter ion groups plays a role in the large red shift of the absorption maximum in the NIR state.

The NIR absorption of NeoR is a quite novel property of rhodopsins and thus significant finding to understand the physicochemical mechanism of the spectral tuning of photoreceptor proteins as well as to develop a new optogenetic tool with an improved light sensitivity in biological tissue. Furthermore, although the functional role of NeoR in the RGC heterodimer is not clear at this moment as the authors admitted, the findings reported in the manuscript could be very interesting if the heterodimerization of RGCs with the optically unique NeoR furnishes the complex with an optical regulation of the light-sensitivity for the cyclase function as the authors speculated. Such a functional regulation would also be of significance for augmentation of the functionality of the optogenetic tools. I therefore consider that the manuscript is suitable for publication after the authors address the following issues.

1. One possible hypothesis of the regulatory role of NeoR may be suppression of a UV sensitivity of the cyclase activity through FRET. It is known that typical microbial rhodopsins exhibit (stimulated) emissions in the NIR region after photoexcitation, which could energetically match the NIR absorption of NeoR. If FRET from RGCs to NeoR can take place, photoexcitation of RGCs by UV light illumination is deactivated through FRET to NeoR in the NIR state which is accumulated by UV light. In contrast, the excitation by light illumination in the visible and NIR regions depletes the NIR FRET acceptor state of NeoR through FRET and direct excitation of NeoR, respectively, which convert the NIR state of NeoR to the UV state, leading to reduction of the deactivation by FRET. It may be interesting to discuss such a hypothesis of FRET involvement. The question is whether FRET between the monomeric units is possible within the short time period in the excited state of RGCs. A theoretical calculation can provide an estimate of the oscillator strength of the excitation from which a rough estimate of the time constant of FRET may be able to be deduced.

2. The QM/MM models of System 5-7 shown in Figure 4 are counter-intuitive in terms of protonation state and structure, although they give NIR absorption maxima consistent with the experimental evidences. Asp140 is protonated even though it forms a hydrogen-bond with the positively charged protonated Schiff base. In System 6, there also are possible protonation states where the other oxygen atoms of the carboxyl group of Glu136 and Glu262 accept protons of which energies could be compared with that of System 6 in the manuscript. The retinal chromophore keeps a highly distorted structure due to the hydrogen-bond even though Asp140 is protonated and thus the interaction with the protonated Schiff base is not very strong. In fact, it looks in Figure 4c for snapshots of the MD simulations that the hydrogen-bond of the protonated Schiff base and the protonated Asp140 is completely dissociated and the former forms a salt bridge with one of the negatively charged deprotonated carboxylates, which is not consistent with the QM/MM models but seems quite reasonable. Given that the use of the atomic structure obtained by the homology modeling is not obvious at all for the present purpose, and the QM/MM optimization strongly depends on the initial structural model due to inevitable trap at a local energy minimum, the authors may wish to discuss

the validity of the QM/MM models more carefully.

Reviewer #2 (Remarks to the Author):

The manuscript by Broser et al. presents their combined experimental/computational work on NeoR, a rhodopsin protein with a characteristic absorption in the near infrared.

Such rhodopsin has never been previously characterized, and no other natural rhodopsin has been, so far, reported as having such a long wavelength absorption, or so high fluorescence yield.

Moreover, the authors prove that this protein is a constituent of hetero complexes with other rhodopsins (another novelty), and it is essential for their function.

Having said this, it is my opinion that the authors fail to raise awareness of the potential importance of such system. In other words, only the very last sentence of the manuscript (lines 208 - 211) states a possible technological use of this rhodopsin or its derivative, without providing enough details to capture the reader's attention. The authors present their new rhodopsin as a curio, without explaining what its importance for the specific field may be, as well as its possible consequences. Furthermore, they fail on addressing the consequences of postulating that such strong red-shift in absorbance, and exceptional fluorescence yield are due only to the particular protonation states of three residues.

Given this, and the major and minor concerns listed below, I consider that the manuscript should not be published in its current form. However, given the appeal of the subject, I would recommend a new submission once all raised concerns have been properly addressed.

Major issues:

A major concern of mine is about the authors claim that their results "revealed the unusual photochemical properties resulting from rigidity of the retinal chromophore and precise spatial organization of the retinal Schiff base, with minimal influence from the remaining retinal-binding pocket." (lines 34 - 36)

I believe there is not enough data presented in the manuscript to substantiate such claim. The authors rightly prepared and experimentally tested a series of variants of NeoR, where they substituted a number of residues supposedly surrounding the retinal chromophore. However, they do not discuss the choice of each residue substitution in terms of expected versus obtained results. In other words, I would have expected that the authors explicitly justify their choice of mutation (i.e., why residue 110 was changed into a Y?) for example, in terms of changed or inverted polarity, changed steric hinderance, etc.

The rigidity claim is, also, not really proven. The authors report that their best MM model that includes, as protonated, only two of the three residues they studied in depth, contained few water molecules close to the retinal moiety. The authors claim that few water molecules imply few degrees of freedom and hence the rigidity claim, as proven by the narrow absorption band. The authors take then a further unproven logical leap, stating that such rigidity is responsible also the extraordinary fluorescence quantum yield. However, the authors may confuse vibrational rigidity that gives rise to narrow absorption bands with molecular rigidity in terms of impossibility to perform the typical photochemical isomerization reaction of a retinal chromophore.

A properly reasoned set of variants, produced to test the plasticity of the cavity, in terms of, e.g., added or removed steric hinderance, along with corresponding experimental photochemical properties and QM/MM models would provide the logical foundation upon which build factual structural claims.

Line 99: the authors claim that NeoR is a bistable rhodopsin, and that illumination with UV light reconstitutes the NIR absorbing state. However, no evidence is provided. Figure 2b shows that, by illuminating for 5 minutes at 680 nm (i.e., close to the absorption maximum of the NIR state), the NIR absorbing species disappears, while a new species appears with an absorption maximum around 367 nm. However, I fail to see any "reversible" event, although so claimed in the figure's caption. There is no timing specified in no parts of the text stating how long does it take for the UV absorbing state to reverse to the starting state. In other words, I see bleaching but I do not see any formation. As an addendum: although relatively common, the authors should have explained the concept of bistable photochromism, at least very briefly.

The chosen methodology for building and employing a QM/MM and later only MM model is not discussed. The authors start by building a homology model of NeoR based on a large series of existing rhodopsins

- how these particular rhodopsins were chosen?
- why so many rhodopsins were necessary?
- what is the advantage of such technique with respect to choosing, i.e., only one rhodopsin with the largest sequence similarity?

A major problem for rhodopsin models as the protonation state of its titratable residues is nearly not addressed. The authors simply report that they applied "Standard protonation states" for all residues, without saying to which standards they refer to. A large literature body is dedicated to the issues arising with different protonation states in rhodopsins, and their effects on their photochemical properties. I would recommend the authors to acquaint themselves with such issues, and thus use them to justify (or change) their choice.

Instead, the authors focus only on three residues close to the protonated Schiff base. While the corresponding study is commendable, I deem not sufficient in terms of studying the electrostatic environment of the retinal cavity, in light of the authors claim that the rest of the residues have little if no influence.

The authors fail to justify their choice of a QM/MM model, in terms of a comparison of results obtained for similar rhodopsins employing their methodology and accuracy of the obtained results against experimental data.

In such sense, the authors should have introduced mutations in their computational model to reflect all the variants that they studied experimentally. A comparison of experimental and computational data would have provided confidence in the model and a proper base for a structural analysis of the cavity residue effects.

The authors should, in general, improve on their literature search. For example, novel Arch rhodopsins have been described, with higher quantum efficiencies than those stated in the manuscript (<https://doi.org/10.1073/pnas.1413987111>). Similarly, when discussing the effects of cavity residues the authors should include at least the following key references:

<https://doi.org/10.1002/chem.201505126> and <https://doi.org/10.1038/srep38425> .

Minor issues:

Line 68: data not shown. This is unacceptable. All data should be provided, at least in the supporting information.

Lines 101 – 104: I fail to see the importance of using 3,4-dehydroretinal, the consequent A2-NeoR, and the corresponding panel in figure 2c.

In the molecular dynamic simulations, the authors clearly employed a fully MM model of NeoR embedded in a membrane. However, they fail to explicitly report which MM force field was employed for the retinal and the protonated linked lysine.

Reviewer #3 (Remarks to the Author):

The work "NeoR, a near-1 infrared absorbing rhodopsin" by M. Broser et al. describes three , RGC1, RGC2, and RGC3 rhodopsin-guanylyl cyclases from *Rhizoclostridium globosum* and additionally functionally characterizes the protein-protein complexes, with a particular focus on the isolated RGC3 (NeoR).

The authors found that the rhodopsin domain of NeoR may absorb the light of the near-infrared region of the spectrum. Particularly, NeoR has an unusually far-red-shifted maximum absorption wavelength of 690 nm, which is the first demonstration of a native rhodopsin with such properties.

The authors also show that NeoR is a photoswitchable bistable protein, where one of the states is highly fluorescent. Importantly, the authors demonstrate high fluorescence of NeoR in ND7/23 cells, which is an important finding. The findings make the protein a unique member of rhodopsins superfamily.

The work by M. Broser et al. contributes to the understanding of the fundamentals of rhodopsins, and also expands the capability of rhodopsins utilization for biotechnological applications, such as, for instance, optogenetics.

The experiments were carefully carried out. A thorough mutational study, together with QM/MM calculations, provides insights into the principles of far-red light absorption and high fluorescence of NeoR.

Undoubtedly, this work is interesting and important. However, it has two parts, poorly connected, which may be misleading and thus confusing for the reader of Nature Communications. Namely, the first part describes the functional characterization of the heterodimers of full-length RGC1/2/3 proteins in living cells. In contrast, the second is mostly focused on the isolated rhodopsin domain of NeoR in detergent. It leads to several questions concerning the biological relevance of the findings.

Indeed, the results of the extensive analysis of NeoR, although being of particular interest, demonstrate the unique properties of almost only the rhodopsin domain of NeoR solubilized in the detergents, but not its complex with RGC1 and RGC2 proteins. Therefore, the relation of this part of the findings to the functional properties of the RGC1/NeoR and RGC2/NeoR heterodimers, described in the first part of the manuscript, is elusive, because of:

1) it is known that the properties of the isolated rhodopsin could differ remarkably from those of a rhodopsin being in the protein complex *in vivo*;

2) the detergent micelles may alter the properties and folding of the rhodopsin, and to avoid this the protein should be reconstituted into a lipid membrane-mimicking system;

and

3) most of the studies were performed using the truncated NeoR protein, while the only spectrum of the full-length protein is provided for NeoR1 from another organism. Although it resembles the shape of the 'red' state of NeoR, neither the bistability nor the fluorescence of the full-length NeoR1 is addressed.

I believe that addressing these issues will not require much work from the authors since they have great experience in the rhodopsin field, and most of the corresponding protocols are already established. At the same time, it will help the authors both to avoid unnecessary speculations and to connect the two parts of the manuscript better and provide a complete picture of the function and spectral properties relationships in RGCs. Taking together, I find the work suitable for publication in Nature Communications, after a major revision addressing the questions concerning the issues listed above:

Minor remarks:

1) Lines 69-71: It is not clear from the description whether RGC2 and NeoR were expressed individually in ND7/23 cells. The statements: "Next, we individually co-expressed RGC1 and RGC2 with NeoR and an engineered cGMP-gated potassium-selective channel in mammalian hybrid cells (ND7/23)"- (lines 63-65) claim that the authors co-expressed RGC1 and RGC2 only with NeoR and the channel, but not alone. The figure on such measurements is also not provided. The authors should

state more clearly that they not only co-expressed the RGC1/NeoR and RGC2/NeoR complexes but also expressed RGC2 and NeoR one by one in the ND7/23 cells and present the results of these experiments.

2) Lines 74-75: It would be useful to quantify the statement on the high conservatism of the cyclase domain of all the three proteins. What is their sequence identity/similarity? An interesting question is whether the putative linkers between the rhodopsin and cyclase domains are conserved or not? The figure with the sequence alignment of the linker regions would be useful for answering this question.

3) Line 78: The statement is not clear. Which of the rhodopsin fragments were expressed in HEK-T and which in Sf21, respectively? What is the reason for the use of different expression systems? The information is important for further studies of RGC1, RGC2, and NeoR and should be provided in the main text.

4) Lines 197-200, 285-286: The authors report that they expressed full-length NeoR1 and RGC1 from *Chytrium confervae* in mammalian cells. However, the authors do not mention the names of the proteins in the main text (lines 197-200). This is confusing, especially the fact that the name of the RGC1 is the same for the proteins from *R. globosum* and *C. confervae*. I suggest the authors give the names of the proteins (NeoR1 and RGC1 from *C. confervae*) directly in the main text. It would also be helpful to rename the RGC1 from *C. confervae* in the same manner as it was made for NeoR protein, to avoid the confusion of the readers.

5) Lines 285-286: Since the authors were able to express and isolate full-length NeoR1 and RGC1 proteins from *C. confervae*, and obtained beautiful spectra of the both (Supplementary Figure 4a), there is a question why other studies were performed with the truncated constructs? The use of full-length construct is more physiologically/biologically relevant. Were the expression and purification protocols the same for the truncated proteins from *R. globosum* and the full-length proteins from *C. confervae*?

6) Supplementary Figures 2 and 3: Please replace NeoRh with NeoR in the graph legend (or directly indicate what NeoRh is) to avoid confusion.

7) Have the authors studied the oligomeric state of NeoR rhodopsin part when purified alone? As I understood from the Materials and Methods section (line 280), they used 100 kDa filter for the concentration of NeoR. It is known that rhodopsins can pass such filters being in monomers, dimers, and even in trimers (for instance, in the case of bacteriorhodopsin). Has NeoR skipped through the 100 kDa filter? Since the conformation and environment of the Schiff base are important for the spectroscopic properties of NeoR, the non-specific oligomerization could alter both the overall protein structure and internal organization, including that of the retinal binding pocket.

8) Related to the previous question: have the authors studied the spectral properties of NeoR in complex with RGC1/2? The co-reconstitution of RGC1/RGC2 with NeoR into lipid vesicles or nanodiscs may give protein complexes, which appear in vivo. Since the proteins are functional being in heterodimers, analysis of the heterodimers is of high importance for the understanding of the mechanisms of protein functioning.

9) An important question is whether there is a biological relevance of the existence of two stable photoswitchable states of NeoR? In other words, which of the states ('red' or 'UV' or both) is considered to provide the functionality of the RGC1/NeoR and RGC2/NeoR complexes and which is not? This problem could be easily addressed by the authors. From the manuscript, I assume that the results of functional studies, shown in Figure 1, relate to the 'red' state of NeoR. I suggest that the electrophysiological studies in *Xenopus* oocytes or ND7/23 cells expressing RGC1/NeoR and RGC2/NeoR complexes should be performed with preliminary (or simultaneous) illumination of the cell by the far-red light to switch the NeoR protein into the 'UV' state. These experiments would answer

whether the complexes are functional with NeoR being in the 'UV' state and would make the manuscript more complete.

10) The authors demonstrate by the QM/MM calculations that the protonation of the Schiff base counterions affects the maximum absorption wavelength in NeoR. They also provide theoretical values for each of the protonation states. It is known that the pH of the surrounding buffer may affect the protonation of the Schiff base counterions. In this case, the spectra measurements at different pH may provide experimental evidence of the calculated findings. I would suggest that the authors should analyze the maximum absorption wavelength of NeoR at different pH values.

11) The authors present a model of the Schiff base region of NeoR, with an unusual D140 residue. D140 is suggested to form a hydrogen bond with the Schiff base. This is an interesting finding. However, D140 is in the similar position as Asp in the NDQ motif of bacterial sodium pumps, for instance, D116 of the KR2 rhodopsin. D116 is also hydrogen-bonded to the Schiff base of KR2 in the ground state but is deprotonated. As the maximum absorption wavelength of KR2 is around 525 nm, I think that this is an additional evidence that D140 in NeoR is protonated and contributes to the rigidity of the chromophore (Figure 4). This should be discussed in the manuscript.

REVIEWER COMMENTS

Reviewer #1 (Remarks to the Author):

The manuscript reports a microbial rhodopsin sensitive to light in the near-infrared (NIR) region called NeoR. The study showed that NeoR forms heteromeric dimers with rhodopsin cyclases, RGC1 and RGC2, respectively, and activates the functions of light-sensitive cyclases in the heterodimeric forms. RGC1 and RGC2 exhibit photo-absorptions centered at 550 nm and 480 nm, respectively. The peak of the absorption spectrum of RGC2 resembles that of the action spectrum, indicating that RGS2 is responsible for the photo-detection for the cyclase function. In contrast, NeoR forms photochromic bi-stable states with absorption maxima at 690 nm and 367 nm, respectively, and the NIR state exhibits a strong fluorescence. Molecular mechanism of the NIR absorption of NeoR was investigated through homology modeling, mutagenesis experiments and theoretical calculations with a hybrid QM/MM method and MD simulations. The study suggested that Asp140 in the vicinity of the protonated Schiff base of the chromophore and one of the counter ion groups plays a role in the large red shift of the absorption maximum in the NIR state.

The NIR absorption of NeoR is a quite novel property of rhodopsins and thus significant finding to understand the physicochemical mechanism of the spectral tuning of photoreceptor proteins as well as to develop a new optogenetic tool with an improved light sensitivity in biological tissue. Furthermore, although the functional role of NeoR in the RGC heterodimer is not clear at this moment as the authors admitted, the findings reported in the manuscript could be very interesting if the heterodimerization of RGCs with the optically unique NeoR furnishes the complex with an optical regulation of the light-sensitivity for the cyclase function as the authors speculated. Such a functional regulation would also be of significance for augmentation of the functionality of the optogenetic tools. I therefore consider that the manuscript is suitable for publication after the authors address the following issues.

1. One possible hypothesis of the regulatory role of NeoR may be suppression of a UV sensitivity of the cyclase

activity through FRET. It is known that typical microbial rhodopsins exhibit (stimulated) emissions in the NIR region after photoexcitation, which could energetically match the NIR absorption of NeoR.

If FRET from RGCs to NeoR can take place, photoexcitation of RGCs by UV light illumination is deactivated through FRET to NeoR in the NIR state which is accumulated by UV light.

In contrast, the excitation by light illumination in the visible and NIR regions depletes the NIR FRET acceptor state of NeoR through FRET and direct excitation of NeoR, respectively, which convert the NIR state of NeoR to the UV state, leading to reduction of the deactivation by FRET. It may be interesting to discuss such a hypothesis of FRET involvement. The question is whether FRET between the monomeric units is possible within the short time period in the excited state of RGCs. A theoretical calculation can provide an estimate of the oscillator strength of the excitation from which a rough estimate of the time constant of FRET may be able to be deduced.

We thank the reviewer for the critical reading of the manuscript and for the valuable comments.

The reviewer's argumentation is in principle correct and an interesting suggestion. But considering the low fluorescence quantum efficiency of the potential donor RGC1 or RGC2 caused by the short excited state lifetime of < 1 ps such an energy transfer would not be expected to contribute substantially to the activity. On the other hand Elena Govorunova has very recently shown that energy transfer occurs from EYFP to a red-shifted ChR "RubyACR" (doi: <https://doi.org/10.1101/2020.04.15.043158>) with the rhodopsin serving as an efficient energy acceptor but in this case the donor EYFP has an excited state lifetime of nanoseconds. We calculated FRET-transfer rates based on an assumed retinal-retinal distance of 20, 25 and 30 Å within the heterodimer with isotropic orientation and a 25 nm blue-shifted BR fluorescence spectrum, that may resemble the unknown spectrum of RGC1 (absorption maximum 550 nm; shown in Supplementary Fig. 10). Our calculations (described in Supplementary Methods) results in FRET lifetimes of 840 fs, 3.2 ps and 9.7 ps. Thus depending on the real, but unknown spatial arrangement of the retinals even a short excited state lifetime may allow a significant energy transfer towards NeoR. Notably, RGC2 has a 70 nm blue-shifted absorption spectrum compared to RGC1 thus a possible regulatory function of FRET should differ between the two functional heterodimers. We performed light-titration experiments of RGC2/NeoR complexes with NeoR either in the (red) or (UV) state archived by continuous background illumination with 660 nm light. While the light saturation behavior reveals unaffected (now in Supplementary Fig. 3a), the photocurrent amplitudes were increased for far-red light treated cells (now shown in main Fig. 1e). The influence of far-red light was further confirmed by measuring the photocurrent before and after a 15 s 660 nm pulse within the same cell (Supplementary Fig. 3b,c). This effect, assigned to NeoR, would match with the functional involvement of FRET as suggested. Unfortunately we were not able to purify the heterodimer yet to obtain experimental evidence for the absence/presence of FRET. We include a short paragraph on a possible FRET effect that could explain the increased photocurrent of RGC2/NeoR₃₆₇ within the discussion section (see page 11, line 258ff.).

2. The QM/MM models of System 5-7 shown in Figure 4 are counter-intuitive in terms of protonation state and structure, although they give NIR absorption maxima consistent with the experimental evidences. Asp140 is protonated even though it forms a hydrogen-bond with the positively charged protonated Schiff base. In System 6, there also are possible protonation states where the other oxygen atoms of the carboxyl group of Glu136 and Glu262 accept protons of which energies could be compared with that of System 6 in the manuscript. The retinal chromophore keeps a highly distorted structure due to the hydrogen-bond even though Asp140 is protonated and thus the interaction with the protonated Schiff base is not very strong. In fact, it looks in Figure 4c for snapshots of the MD simulations that the hydrogen-bond of the protonated Schiff base and the protonated Asp140 is completely dissociated and the former forms a salt bridge with one of the negatively charged deprotonated carboxylates, which is not consistent with the QM/MM models but seems quite reasonable. Given that the use of the atomic structure obtained by the homology modeling is not obvious at all for the present purpose, and the QM/MM optimization strongly depends on the initial structural model due to inevitable trap at a local energy minimum, the authors may wish to discuss the validity of the QM/MM models more carefully.

The two types of simulations and their results have different accuracy and different purpose.

The QM/MM geometry optimization in panel 4b shows the final relaxed geometry that was started from the homology model. In the QM/MM simulation, we have chosen a large QM region which includes the retinal PSB, lysine and 3 carboxyl groups. This large choice of the QM region allows describing bond breaking and bond formation if the energy can be lowered, for example in the case of proton transfer. In case of system 6, this is indeed what we have observed. The large QM region also account for polarization between the groups that are included, hence the charges of the oxygen atoms in a carboxylate can be different dependent on the local environment or interaction partner. All these effects (as now stated on page 9, line 212ff. of the manuscript) are not available in the MM region which is described by a static force field, e.g. the oxygen in a carboxylate have always the same charge, independent of the specific environment.

The goal of the QM/MM geometry optimization is to obtain the best possible geometry (lowest in energy) for the calculation of excitation energies, because, as the reviewer correctly wrote, the initial structure from the homology modelling is not reliable and needs to be refined. Regarding the Asp140 hydrogen bonding, the oxygen has a lone pair which is interacting with the proton of the Schiff base. In order to show that Asp140 indeed binds to PSB+ we have analyzed the noncovalent interactions by generating the reduced density gradient (RDG)

isosurfaces at B3LYP/6-31+G(d,p) level of theory. The Figure below shows the RDG including the one between D140 and PSB.

Figure. Intermolecular reduced density gradient (RDG) isosurface graphs for the noncovalent interactions. The colors of interactions, (1) blue: stronger noncovalent interactions; (2) green-light brown: weaker noncovalent interactions. The analyses were carried out using the Multiwfn package and the isosurface graphs were generated using the VMD program.

The reviewer is right; if we would run a long molecular dynamics simulation using this QM/MM partitioning we would probably see a dynamic proton network with proton transfer between different carboxylates. However, such a simulation is not feasible at the QM level of theory. But since our results show that the same charge of the QM region gives similar excitation energies, we can make our conclusion general and deduce that system 5 and 6/7 are responsible for the strong red shift.

The MD simulation covers a long time range of 300 ns and is performed by describing the entire NeoR embedded in the membrane at the MM level. Hence, the effects included in the QM/MM simulation are missing. However, the purpose of this type of simulation is to test the stability of the homology model and to see how much water can penetrate in the retinal binding pocket. System 8 of Fig. 4a, for example, was not stable. Panel c shows the density of the water from the entire MD trajectory. In order to indicate the relative orientation to the retinal PSB, we have also shown the geometry of retinal in the homology model, which was the initial structure in the MD simulation. The same structure was used in the QM/MM geometry optimization as a starting point. Hence, it shows that the QM/MM optimization can be different from the initial geometry,

Nevertheless, following the request of the reviewer, we now include an alternate calculation with a smaller QM region comprising the protonated retinal Schiff base and the side chain of K266. Following the suggestion of the reviewer we protonated each carboxylate oxygen and generated initial guesses with both forms. We minimized each of these initial guess geometries and chose the lowest-energy structure for the computation of excitation energies. These results are now slightly blue-shifted by 0.1–0.2 eV however, but are still within the accuracy limit of the QM/MM method (the difference looks large in wavelengths which are not proportional in energy). By taking the counterions out of the QM region, we prevent their polarization and restrict each respective proton to its predefined position. And as expected these models are more consistent with the hydrogen-bonding networks observed during the MD simulation, i.e. the Schiff base forms a hydrogen-bond with one of the deprotonated counterions. Qualitatively, we still see the same trend, where increasing the amount of protonated counterions causes a red shift (see Supplementary Fig.8 and Supplementary Methods). Decreasing the size of the QM region did not provide us additional insights, yet it further strengthened the previously determined effect of the counterion protonation. Moreover, exclusion of the counterions from the QM region actually decreased the quality of the computed excitation energies, and highlighted the importance of the QM interactions between retinal and the counterions.

Reviewer #2 (Remarks to the Author):

The manuscript by Broser et al. presents their combined experimental/computational work on NeoR, a rhodopsin protein with a characteristic absorption in the near infrared.

Such rhodopsin has never been previously characterized, and no other natural rhodopsin has been, so far, reported as having such a long wavelength absorption, or so high fluorescence yield. Moreover, the authors prove that this protein is a constituent of hetero complexes with other rhodopsins (another novelty), and it is essential for their function.

Having said this, it is my opinion that the authors fail to raise awareness of the potential importance of such system. In other words, only the very last sentence of the manuscript (lines 208 - 211) states a possible technological use of this rhodopsin or its derivative, without providing enough details to capture the reader's attention. The authors present their new rhodopsin as a curio, without explaining what its importance for the specific field may be, as well as its possible consequences. Furthermore, they fail on addressing the consequences of postulating that such strong red-shift in absorbance, and exceptional fluorescence yield are due only to the particular protonation states of three residues.

We are certainly grateful for such a comment but we hesitate to mention any application too explicitly. Our experience is that any suggestion for application triggers the reaction of the reviewer to prove application right away. However, we expand the paragraph about the impact of infrared-absorbing photoreceptor proteins on possible optogenetic applications. The present work reports the discovery of a new rhodopsin with unmet properties, its characterization by spectroscopy, electrophysiology and computer simulations. We are planning a follow-up where we can address the photochemical properties in detail. We hope that the reviewer can appreciate our combined effort and the reported discovery in this manuscript.

Given this, and the major and minor concerns listed below, I consider that the manuscript should not be published in its current form. However, given the appeal of the subject, I would recommend a new submission once all raised concerns have been properly addressed.

Major issues:

A major concern of mine is about the authors claim that their results "revealed the unusual photochemical properties resulting from rigidity of the retinal chromophore and precise spatial organization of the retinal Schiff base, with minimal influence from the remaining retinal-binding pocket." (lines 34 – 36)

I believe there is not enough data presented in the manuscript to substantiate such claim. The authors rightly prepared and experimentally tested a series of variants of NeoR, where they substituted a number of residues supposedly surrounding the retinal chromophore. However, they do not discuss the choice of each residue substitution in terms of expected versus obtained results.

We do not exclude that modification of the retinal binding pocket contributes to the red-shifted absorption, but the mutation of all residues within 5 Å distance except the active site have a small influence which is not larger and in several cases even smaller than for example in Bacteriorhodopsin (see Supplementary Table1) but by no means they are responsible for the extra 120 nm red-shift. This extra shift is exclusively caused by the unusual active site configuration, which we now expressed even more explicitly.

In other words, I would have expected that the authors explicitly justify their choice of mutation (i.e., why residue 110 was changed into a Y?) for example, in terms of changed or inverted polarity, changed steric hinderance, etc.

We may not have done it in the original version well enough in order not to extend the text too much. We mainly focused on polarity changes but if possible with otherwise moderate perturbation, and we focused on substitutions that have been done for Bacteriorhodopsin - which is quite red shifted already compared to other microbial rhodopsins - in pioneering work mainly of the late Khorana group. The related BR mutations together with their spectral impact are also listed in Supplementary Table1. Certainly, we can do more but owing to the countless combinations it would be an almost endless business. Nevertheless, the reviewer is correct in his comment and we now included a more extended justification for our choices (see page 7, line 174ff.). W110Y for example was chosen because in BR the homolog position is occupied by Y and the OH group of the Tyr should be close to the NH-group of the Trp.

The rigidity claim is, also, not really proven. The authors report that their best MM model that includes, as protonated, only two of the three residues they studied in depth, contained few water molecules close to the retinal moiety. The authors claim that few water molecules imply few degrees of freedom and hence the rigidity claim, as proven by the narrow absorption band. The authors take then a further unproven logical leap, stating that such rigidity is responsible also the extraordinary fluorescence quantum yield. However, the authors may confuse vibrational rigidity that gives rise to narrow absorption bands with molecular rigidity in terms of impossibility to perform the typical photochemical isomerization reaction of a retinal chromophore. A properly reasoned set of variants, produced to test the plasticity of the cavity, in terms of, e.g., added or removed steric hinderance, along with corresponding experimental photochemical properties and QM/MM models would provide the logical foundation upon which build factual structural claims.

We partially agree that the rigidity of the NeoR chromophore cannot be derived directly from the theoretical models. But we would like to point out the experimental evidence from the absorption/fluorescence spectra: a) due to the fact that the emission band of NeoR is a mirror image of the absorption band, it is clear that the change in the geometry upon excitation is small. b) from the small Stokes shift (0.043 eV) we can conclude that the relaxation energy (λ) is also small. These two facts, along with the existence of the vibrational fine structure in both the absorption and emission bands, allow us to conclude that the Frank-Condon point is located near the minimum of the excited state and has not enough energy to overcome the potential barrier to undergo the photoisomerization. This explains the high fluorescence quantum yield of 20% (compared to the one in Bacteriorhodopsin: 0.0025% – 0.027%). Thus based on these observations we expect the NeoR chromophore to be rigid. However, for those theoretical models with one carboxylate deprotonated we found red-shifted absorption energies by QM/MM that match the experimentally observed spectra and, in addition, the MD simulation show low water content close to the RSB that would explain the narrow spectral shape observed for NeoR. In such a way both theoretical approaches support our hypothesis that only one of the three carboxylates of the NeoR active site is deprotonated. We now rephrased the respective paragraphs regarding our conclusions about the rigidity of the NeoR chromophore (see page 6, line 129ff.).

Line 99: the authors claim that NeoR is a bistable rhodopsin, and that illumination with UV light reconstitutes the NIR absorbing state. However, no evidence is provided. Figure 2b shows that, by illuminating for 5 minutes at 680 nm (i.e., close to the absorption maximum of the NIR state), the NIR absorbing species disappears, while a new species appears with an absorption maximum around 367 nm. However, I fail to see any “reversible” event, although so claimed in the figure’s caption. There is no timing specified in no parts of the text stating how long does it take for the UV absorbing state to reverse to the starting state. In other words, I see bleaching but I do not see any formation.

As an addendum: although relatively common, the authors should have explained the concept of bistable photochromism, at least very briefly.

Data about reversible bleaching of detergent purified NeoR rhodopsin fragment, as well as detergent purified full-length NeoR and the NeoR-rhodopsin fragment embedded into SMA protein/lipid nanodiscs are now included in Fig. 2 b-d and Supplementary Fig. 5. Further we briefly introduce the concept of bistable photochromism on page 5, line 109.

The chosen methodology for building and employing a QM/MM and later only MM model is not discussed. The authors start by building a homology model of NeoR based on a large series of existing rhodopsins

- how these particular rhodopsins were chosen?
- why so many rhodopsins were necessary?
- what is the advantage of such technique with respect to choosing, i.e., only one rhodopsin with the largest sequence similarity?

The rhodopsins for the homology modeling were chosen by searching the Protein Data Bank for the retinal proteins with the highest sequence agreement. All structures had a sequence identity below 20% therefore we had to combine several templates to compensate for the lack of a single protein structure with sufficient similarity. We wanted to avoid biasing our model too much by restricting ourselves, e.g. only to bacteriorhodopsin and generating a homology model that would be like this template. So we chose crystal structures of different proteins with a sequence identity between 10% and 17% as now described on page 7, line 167ff.

A major problem for rhodopsin models as the protonation state of its titratable residues is nearly not addressed. The authors simply report that they applied “Standard protonation states” for all residues, without saying to which standards they refer to. A large literature body is dedicated to the issues arising with different protonation states in rhodopsins, and their effects on their photochemical properties. I would recommend the authors to acquaint themselves with such issues, and thus use them to justify (or change) their choice. Instead, the authors focus only on three residues close to the protonated Schiff base. While the corresponding study is commendable, I deem not sufficient in terms of studying the electrostatic environment of the retinal cavity, in light of the authors claim that the rest of the residues have little if no influence.

We are aware of the broad literature about computational studies which treat protonation states in rhodopsins, but these studies are based on the structural information arising from crystallography and none of them is dealing with an opsin shift comparable to NeoR. Since the photochemical properties of NeoR are unique and the structural data is limited we have selected the counterions E136, D140 and E262 because our experimental studies had shown these to have by far the biggest spectral impact. These residues are the most likely candidates for the spectral tuning because they are next to the Schiff base and because there are no other titratable residues in the binding pocket as defined by 5 Å distance. Therefore we applied the standard protonation state at pH 7 for the remaining titratable residues, as used as reference in MD simulations, i.e. a glutamate would remain deprotonated. The current study mainly reports the discovery and functional/biophysical characterization of NeoR as the first described NIR absorbing rhodopsin. Our theoretical part of the study is focused exclusively on the possible organization of the three carboxylates of the counterion complex, in particular their protonation pattern.

Guided by the experimental evidence, these three carboxylates define the far-red absorption in NeoR, but the QM/MM models and the MD simulation indicating that most likely only one of the carboxylates is protonated. The little influence of the remaining retinal cavity to the extreme opsin shift was deduced from the mutation study.

The authors fail to justify their choice of a QM/MM model, in terms of a comparison of results obtained for similar rhodopsins employing their methodology and accuracy of the obtained results against experimental data. In such sense, the authors should have introduced mutations in their computational model to reflect all the variants that they studied experimentally. A comparison of experimental and computational data would have provided confidence in the model and a proper base for a structural analysis of the cavity residue effects.

In order to demonstrate the accuracy of our QM/MM setup and to address the request of the reviewer we have also calculated bacteriorhodopsin and the light driven sodium pump KR2 (which also has D140 as NeoR) using the same protocol, namely using the ADC(2) method for QM/MM calculations. The details of these calculations are available in the Supplementary Methods section. The results for BR and KR2 are 2.50 eV and 2.61 eV, respectively (as shown now in Supplementary Fig. 8). Hence, the excitation energy is systematically underestimating the experiment counterpart and gives us more confidence in the accuracy of our protocol. The papers that the reviewer is citing below about mutations within the retinal cavity are based on rhodopsin mimics where crystal structures are available, in most cases even for mutants, because they are soluble proteins and not membrane proteins. However, despite our effort to construct the best possible model, we are limited by the lack of structural information. In this sense we employed QM/MM in order to compare the effect of different active site configuration, rather than to unravel the detailed contribution of single amino acids.

The authors should, in general, improve on their literature search. For example, novel Arch rhodopsins have been described, with higher quantum efficiencies than those stated in the manuscript (<https://doi.org/10.1073/pnas.1413987111>).

We totally agree with the reviewer that this is a wonderful publication and we studied it in detail. We had included a discussion of this publication and the very red-shifted engineered Retinol binding protein hCRBP II in a previous version and reintroduced it now (see on page 8, line 196ff. and page 9, line 201ff.). We left the hCRBP II discussion out before because it is a semi-artificial soluble protein with almost no relation to rhodopsins and no counterions, but we agree that the comparison will be useful.

Now we also included the mentioned Ref. Mclsaac et al. 2014 (see Ref. #21) and are referring to the brightest engineered Arch-7 with a fluorescence quantum efficiency of 1.2%.

Similarly, when discussing the effects of cavity residues the authors should include at least the following key references: <https://doi.org/10.1002/chem.201505126> and <https://doi.org/10.1038/srep38425>.

We included the suggested Ref. Suomivuori et al. 2016 in our paragraph about hCRBP II (see Ref. #37). The Luk et al. 2016 is not very appropriate for us because the opsin shifts of the discussed rhodopsins are between 44 and 76 nm, which are tiny shifts compared to NeoR where we have to explain a 250 nm shift. Moreover, the Luk et al. paper is discussing 11-cis retinal chromophores, where the situation is quite different. The focus of the publication is on the relation between wavelength and thermal isomerization and noise. It is a great publication but not useful for us at the moment. Nevertheless, we are briefly referring to it (see Ref. #15).

Minor issues:

Line 68: data not shown. This is unacceptable. All data should be provided, at least in the supporting information.

We show the localization of RGC1/NeoR and RGC2/NeoR in ND7/23 cells as well as reversible bleaching of the NeoR fluorescence now in Supplementary Fig.4.

Lines 101 – 104: I fail to see the importance of using 3,4-dehydroretinal, the consequent A2-NeoR, and the corresponding panel in figure 2c.

Here we have a different opinion. On one hand, this experiment plastically demonstrates that the retinal moiety is indeed the red-absorbing chromophore and the absorption difference between A1 and A2-NeoR is the largest shift ever produced in a retinal protein to our knowledge (as we point out now on page 5, line 122,123).

In the molecular dynamic simulations, the authors clearly employed a fully MM model of NeoR embedded in a membrane. However, they fail to explicitly report which MM force field was employed for the retinal and the protonated linked lysine.

For all the simulations we have used the CHARMM force field, including the retinal parameters.

Reviewer #3 (Remarks to the Author):

The work “NeoR, a near-1 infrared absorbing rhodopsin” by M. Broser et al. describes three , RGC1, RGC2, and RGC3 rhodopsin-guanylyl cyclases from *Rhizoclostratium globosum* and additionally functionally characterizes the protein-protein complexes, with a particular focus on the isolated RGC3 (NeoR).

The authors found that the rhodopsin domain of NeoR may absorb the light of the near-infrared region of the spectrum. Particularly, NeoR has an unusually far-red-shifted maximum absorption wavelength of 690 nm, which is the first demonstration of a native rhodopsin with such properties.

The authors also show that NeoR is a photoswitchable bistable protein, where one of the states is highly fluorescent. Importantly, the authors demonstrate high fluorescence of NeoR in ND7/23 cells, which is an important finding. The findings make the protein a unique member of rhodopsins superfamily.

The work by M. Broser et al. contributes to the understanding of the fundamentals of rhodopsins, and also expands the capability of rhodopsins utilization for biotechnological applications, such as, for instance, optogenetics.

The experiments were carefully carried out. A thorough mutational study, together with QM/MM calculations, provides insights into the principles of far-red light absorption and high fluorescence of NeoR.

Undoubtedly, this work is interesting and important. However, it has two parts, poorly connected, which may be misleading and thus confusing for the reader of Nature Communications. Namely, the first part describes the functional characterization of the heterodimers of full-length RGC1/2/3 proteins in living cells. In contrast, the second is mostly focused on the isolated rhodopsin domain of NeoR in detergent. It leads to several questions concerning the biological relevance of the findings.

First of all we are grateful to the reviewer for the critical but positive comments on our manuscript. However, the reviewer is concerned about the disconnected parts one and two. Our reason for the organization of the manuscript in these two really existing parts is the following. Part one describes the discovery of heterodimeric rhodopsin never described before and, more specifically a heterodimeric Rhodopsin-cyclase and we have demonstrated that it enzymatically functionally and the function is controlled by light. The next observation is that dimeric cyclase is activated by the green or blue absorbing monomer although one of the monomers is connected to the NIR-absorbing monomer. During the review process we completed our electrophysiological measurements on RGC2/NeoR in ND7/23 cells that indicate that far-red light is modulating the photocurrent amplitudes, evoked by blue light flashes (see main Fig. 1e, Supplementary Fig.3a-c). We assigned the observed effect to NeoR, which now provides a physiological role for NeoR and allows us to better connect the two parts as criticized by the reviewer. Nevertheless, the spectral properties of the NeoR are the most spectacular aspect of the heterodimeric RGCs and – to our judgement very consequently – we strongly focused in the following on the NeoR properties.

Indeed, the results of the extensive analysis of NeoR, although being of particular interest, demonstrate the unique properties of almost only the rhodopsin domain of NeoR solubilized in the detergents, but not its complex with RGC1 and RGC2 proteins. Therefore, the relation of this part of the findings to the functional properties of the RGC1/NeoR and RGC2/NeoR heterodimers, described in the first part of the manuscript, is elusive, because of:

- 1) it is known that the properties of the isolated rhodopsin could differ remarkably from those of a rhodopsin being in the protein complex *in vivo*;
- 2) the detergent micelles may alter the properties and folding of the rhodopsin, and to avoid this the protein should be reconstituted into a lipid membrane-mimicking system; and
- 3) most of the studies were performed using the truncated NeoR protein, while the only spectrum of the full-length protein is provided for NeoR1 from another organism. Although it resembles the shape of the ‘red’ state of NeoR, neither the bistability nor the fluorescence of the full-length NeoR1 is addressed.

So far it has not been feasible for us to purify functional heterodimers to investigate the possible influence of heterodimerisation on the NeoR photochemistry. However we now included data from fluorescence microscopy on RGC1/NeoR and RGC2/NeoR heterodimers in ND7/23 cells that show co-localization of RGC1/2 with NeoR and reversible bleaching of NeoR fluorescence (Supplementary Fig.4a,b). The latter resemble the features of NeoR full-length protein expressed alone (Fig. 2h). Therefore we conclude that the NeoR photochemistry should in principle remain similar in the heterodimeric complexes.

Regarding point 2, we only partially agree with reviewer in this point. Most of the many rhodopsins that we analyzed over the last 30 years exhibited properties in detergent very similar to those in membranes. However, the absorption may be slightly shifted (< 20 nm), the pK_A of the RSBH⁺ could be modified in detergent and kinetics of the photocycle agreed in most cases only within a factor of two. But, in line with the reviewer, proteins are generally more stable in a lipid environment and in case of the homodimeric Rhodopsin-cyclases the dark activity was reduced. Thus we include spectroscopic data on the rhodopsin-domain of NeoR in styrene maleic acid copolymer lipid particles (SMALPS) to exclude spectral artifacts due to the detergent treatment. The photochemical properties of NeoR in SMALPS remain very similar to the spectral features obtained in detergent micelles, albeit the absorption maximum was shifted to 686 nm and the kinetic of photoconversion was slowed down (shown Supplementary Fig. 5c). Further, we repeated reversible bleaching with full-length NeoR from *Rhizoclostratium globosum* in detergent to show the similar photochemistry of full-length NeoR and prepare a comprehensive overview of all three sample types in Supplementary Fig. 5. Further we now include our preliminary observations about the similar photochemical and functional properties of CcNeoR1 (in Supplementary Fig. 9).

I believe that addressing these issues will not require much work from the authors since they have great experience in the rhodopsin field, and most of the corresponding protocols are already established. At the same time, it will help the authors both to avoid unnecessary speculations and to connect the two parts of the manuscript better and provide a complete picture of the function and spectral properties relationships in RGCs.

We did as suggested.

Taking together, I find the work suitable for publication in Nature Communications, after a major revision addressing the questions concerning the issues listed above:

Minor remarks:

1) Lines 69-71: It is not clear from the description whether RGC2 and NeoR were expressed individually in ND7/23 cells.

The statements: "Next, we individually co-expressed RGC1 and RGC2 with NeoR and an engineered cGMP-gated potassium-selective channel in mammalian hybrid cells (ND7/23)"- (lines 63-65) claim that the authors co-expressed RGC1 and RGC2 only with NeoR and the channel, but not alone. The figure on such measurements is also not provided. The authors should state more clearly that they not only co-expressed the RGC1/NeoR and RGC2/NeoR complexes but also expressed RGC2 and NeoR one by one in the ND7/23 cells and present the results of these experiments.

We tested single expression of RGC2 and NeoR on three cells per construct but no current was observed. We rephrased the paragraph to point out that RGC2 and NeoR has been expressed in ND7/23 cells also one by one (page 3, line 70ff.).

2) Lines 74-75: It would be useful to quantify the statement on the high conservatism of the cyclase domain of all the three proteins. What is their sequence identity/similarity? An interesting question is whether the putative linkers between the rhodopsin and cyclase domains are conserved or not? The figure with the sequence alignment of the linker regions would be useful for answering this question.

We include a multiple sequence alignment of the linker and the cyclase region in Supplementary Fig. 2. and specify the degree of conservation in the figure legend.

3) Line 78: The statement is not clear. Which of the rhodopsin fragments were expressed in HEK-T and which in Sf21, respectively? What is the reason for the use of different expression systems? The information is important for further studies of RGC1, RGC2, and NeoR and should be provided in the main text.

Indeed we used both systems for all three proteins and it turned out that NeoR expresses well in both of the systems. However, the expression of RGC1 and RGC2 rhodopsin fragment is low irrespective of the system used. But, an acceptable spectrum of RGC1 was obtained from HEK-T and that of RGC2 from a sample expressed in Sf21. We now included this information in the text (page 4, line 86ff.). HEK-T cell expression is fast and these cells were used for all mutants but the amount of material we get is small. For spectroscopic studies especially pump-probe experiments we need more material only yielded from Sf21 cells.

4) Lines 197-200, 285-286: The authors report that they expressed full-length NeoR1 and RGC1 from *Chytriomycetes confervae* in mammalian cells. However, the authors do not mention the names of the proteins in the main text (lines 197-200). This is confusing, especially the fact that the name of the RGC1 is the same for the proteins from *R. globosum* and *C. confervae*. I suggest the authors give the names of the proteins (NeoR1 and RGC1 from *C. confervae*) directly in the main text. It would also be helpful to rename the RGC1 from *C. confervae* in the same manner as it was made for NeoR protein, to avoid the confusion of the readers.

We fully agree and have done it accordingly

5) Lines 285-286: Since the authors were able to express and isolate full-length NeoR1 and RGC1 proteins from *C. confervae*, and obtained beautiful spectra of the both (Supplementary Figure 4a), there is a question why other studies were performed with the truncated constructs? The use of full-length construct is more physiologically/biologically relevant. Were the expression and purification protocols the same for the truncated proteins from *R. globosum* and the full-length proteins from *C. confervae*?

The reason to work with the rhodopsin fragment is the low solubilization of full-length NeoR by detergent (the same we observed for other enzyme-rhodopsins), that rarely result enough purified material. The two shown proteins from *C. confervae* were solubilized and purified using the protocol as for the truncated rhodopsin domains, which reveal just enough material for a single spectrum. Therefore, all our mutants as well as the spectra of RGC1/2 have been done with the truncated rhodopsin fragment, nevertheless full-length NeoR from Sf21 cells now allowed us to repeat the key experiments as shown now in Supplementary Fig.5.

6) Supplementary Figures 2 and 3: Please replace NeoRh with NeoR in the graph legend (or directly indicate what NeoRh is) to avoid confusion.

Ok, done.

7) Have the authors studied the oligomeric state of NeoR rhodopsin part when purified alone? As I understood from the Materials and Methods section (line 280), they used 100 kDa filter for the concentration of NeoR. It is known that rhodopsins can pass such filters being in monomers, dimers, and even in trimers (for instance, in the case of bacteriorhodopsin). Has NeoR skipped through the 100 kDa filter? Since the conformation and environment of the Schiff base are important for the spectroscopic properties of NeoR, the non-specific oligomerization could alter both the overall protein structure and internal organization, including that of the retinal binding pocket.

From our experience, the most critical parameter regarding the choice of the concentrator MWCO is the detergent used for purification. In case of DDM/CHS we found that even empty detergent micelles do not pass a 100 kDa filter, and indeed the NeoR rhodopsin fragment could be concentrated quantitatively. However from the gel filtration profile we do see a minor portion of NeoR that may resemble non-specific oligomers, but most of the protein seems to adapt an oligomeric state that result in a similar retention volume as we found for anion-conducting channelrhodopsin GtACR1 with known homodimeric structure. However the precise determination of the oligomeric state of membrane proteins by this approach reveals difficult as the outcome is notoriously influenced e.g. by the shape of the detergent-belt. Therefore we would not conclude from this observation that NeoR (in absence of RGC1/2) forms homodimers. Nevertheless, we found no spectroscopic difference between different fractions of the gel filtration.

8) Related to the previous question: have the authors studied the spectral properties of NeoR in complex with RGC1/2? The co-reconstitution of RGC1/RGC2 with NeoR into lipid vesicles or nanodiscs may give protein complexes, which appear *in vivo*. Since the proteins are functional being in heterodimers, analysis of the heterodimers is of high importance for the understanding of the mechanisms of protein functioning.

We fully agree with the reviewer that this would be an interesting experiment but the yield of RGC1 and RGC2 were not high enough to attempt the reconstitution of heterodimers e.g. in lipid vesicles. Nevertheless, we could follow the bleaching and recovery of NeoR fluorescence in ND7/23 and HEK-T cells co-transfected with both proteins. From this we conclude that the spectral properties of NeoR within the heterodimeric complexes *in vivo* resemble in principle similar photochemistry as the *in vitro* monomers or homomultimers.

9) An important question is whether there is a biological relevance of the existence of two stable photoswitchable states of NeoR? In other words, which of the states ('red' or 'UV' or both) is considered to provide the functionality of the RGC1/NeoR and RGC2/NeoR complexes and which is not? This problem could be easily addressed by the authors. From the manuscript, I assume that the results of functional studies, shown in Figure 1, relate to the 'red' state of NeoR. I suggest that the electrophysiological studies in *Xenopus* oocytes or ND7/23 cells expressing RGC1/NeoR and RGC2/NeoR complexes should be performed with preliminary (or simultaneous) illumination of the cell by the far-red light to switch the NeoR protein into the 'UV' state. These experiments would answer whether the complexes are functional with NeoR being in the 'UV' state and would make the manuscript more complete.

Certainly this is a fully logic and important suggestion. We know that green or blue light is effective for RGC catalytic activation, but when we tested various far-red illumination protocols for TEVC measurements in *Xenopus* oocytes, we could not detect any significant changes in photoresponse. However, the signal variance in oocytes is very high, as seen in Figure 1c and repetitive measurements on single cells are not feasible, making a conclusive analysis of gradual differences based on the state of NeoR difficult. The situation is much better for whole cell voltage clamp measurements in ND7/23 cells, we therefore continued working with this system. Our new co-author Anika Spreen performed light-titration experiments with and without far-red (660 nm) background illumination and observed that the photocurrents evoked by 490 nm light flashes are increased (shown in Fig. 2e), while the light intensity dependence of the signal remains unaffected (shown in Supplementary Fig. 3a). She then verified this effect within single cell measurements by applying a 15 s 660 nm pulse to compare the photoresponse before and after red-light illumination. By this approach we found again that red-light pre-treatment significantly increased the photocurrent (shown in Supplementary Fig. 3 b,c). Unfortunately, we were not able to reverse the process by applying 365 nm light in a clear manner, because the high UV light dose needed for reconversion strongly activated the enzyme (in contrast to 660 nm light, that evoke no response as shown in Supplementary Fig. 3c) and lowered the patch parameters. We assigned the increase of photocurrents upon red-light illumination to the function of NeoR and discussed possible mechanism. We think that these new data make the most substantial improvement of the manuscript now because they unravel a function for NeoR in the heterodimeric complex.

10) The authors demonstrate by the QM/MM calculations that the protonation of the Schiff base counterions affects the maximum absorption wavelength in NeoR. They also provide theoretical values for each of the protonation states. It is known that the pH of the surrounding buffer may affect the protonation of the Schiff base counterions. In this case, the spectra measurements at different pH may provide experimental evidence of the calculated findings. I would suggest that the authors should analyze the maximum absorption wavelength of NeoR at different pH values.

This is a very rational conclusion. We performed pH-Titration and found that the far-red species disappears with a pK_a of ~ 9.3 , giving rise to a species at ~ 410 nm. However the RSBH⁺ obviously is efficiently shielded and the assignment of the newly evolved state one of the calculated protonation pattern is elusive. The observed process is irreversible and may resemble collapse of the retinal binding pocket due to protein denaturation.

11) The authors present a model of the Schiff base region of NeoR, with an unusual D140 residue. D140 is suggested to form a hydrogen bond with the Schiff base. This is an interesting finding. However, D140 is in the similar position as Asp in the NDQ motif of bacterial sodium pumps, for instance, D116 of the KR2 rhodopsin. D116 is also hydrogen-bonded to the Schiff base of KR2 in the ground state but is deprotonated. As the maximum absorption wavelength of KR2 is around 525 nm, I think that this is an additional evidence that D140 in NeoR is protonated and contributes to the rigidity of the chromophore (Figure 4). This should be discussed in the manuscript.

The situation in KR2 is different in the sense that D140 homolog (D116 in KR2) is not connected (or only weakly coupled via water molecules) with the E262 (D251 in KR2) because the second classic counterion residue E136 (N112) is missing in KR2. Therefore, we decide not to include a comparison with KR2 that we think will confuse the general reader. However we included a comprehensive QM/MM calculation on KR2 and BR that show the expected higher excited state energy for these systems in the Supplement in order to validate our QM/MM approach.

REVIEWERS' COMMENTS

Reviewer #1 (Remarks to the Author):

I see that the authors have reasonably addressed the reviewer's comments in the revised manuscript. I therefore believe that the manuscript is now suitable for publication.

Reviewer #2 (Remarks to the Author):

Review of the manuscript titled:

"NeoR, a near-infrared absorbing rhodopsin"

by Matthias Broser, Anika Spreen, Patrick E. Konold, Enrico Peter, Suliman Adam, Veniamin Borin, Igor Schapiro, Reinhard Seifert, John T.M. Kennis, Yinth Andrea Bernal Sierra, and Peter Hegemann

The revised version of this manuscript represents a great leap in terms of clarity and provided information. I really commend the authors for their revision work.

I appreciate their answers to my comments, as well as those of the other reviewers, and I can empathize when they complain of not having previously specified about possible technological usage for fear of being asked or rejected because they do not show such uses. I think the current version provides a good equilibrium between showing possible application, to underline the importance of their system, and not going into details of what these technological uses could be.

Overall, the manuscript is now more cohesive and provides a more incisive story, which I deem worthy of being published in Nature Communications.

I believe I have only one minor concern left, and it is still related to the protonation states of the employed QM/MM model. I understand that the authors have to use a homology model, since there is no crystallographic structure available, and I understand that the outstanding red-shift of the absorbance is, likely, due to strong effects onto the retinal, i.e., originated close by.

Nevertheless, other titratable amino acid may still influence the absorbance. I would recommend the authors to check the ionization states of their homology model with a program such as PROPKA, instead of assigning them as simply as those found at pH 7.0. Possibly, nothing would change in the results, but they would strengthen their argument for the importance of the specific ionization states of the triad close by the protonated retinal Schiff base.

Reviewer #3 (Remarks to the Author):

The authors properly addressed my comments in the revised manuscript.

I would suggest that the revised manuscript is suitable for publication in Nature Communications

Dear Dr. Mieck,

Please find below our response regarding the remaining concern of Reviewer #2.

Best regards,

Matthias Broser

Reviewer #2 (Remarks to the Author):

Review of the manuscript titled:

“NeoR, a near-infrared absorbing rhodopsin”

by Matthias Broser, Anika Spreen, Patrick E. Konold, Enrico Peter, Suliman Adam, Veniamin Borin, Igor Schapiro, Reinhard Seifert, John T.M. Kennis, Yinth Andrea Bernal Sierra, and Peter Hegemann

The revised version of this manuscript represents a great leap in terms of clarity and provided information. I really commend the authors for their revision work.

I appreciate their answers to my comments, as well as those of the other reviewers, and I can empathize when they complain of not having previously specified about possible technological usage for fear of being asked or rejected because they do not show such uses. I think the current version provides a good equilibrium between showing possible application, to underline the importance of their system, and not going into details of what these technological uses could be.

Overall, the manuscript is now more cohesive and provides a more incisive story, which I deem worthy of being published in Nature Communications.

I believe I have only one minor concern left, and it is still related to the protonation states of the employed QM/MM model. I understand that the authors have to use a homology model, since there is no crystallographic structure available, and I understand that the outstanding red-shift of the absorbance is, likely, due to strong effects onto the retinal, i.e., originated close by.

Nevertheless, other titratable amino acid may still influence the absorbance. I would recommend the authors to check the ionization states of their homology model with a program such as PROPKA, instead of assigning them as simply as those found at pH 7.0. Possibly, nothing would change in the results, but they would strengthen their argument for the importance of the specific ionization states of the triad close by the protonated retinal Schiff base.

We have computed the pKa for our homology model using PROPKA3. The results of the PROPKA calculation agrees with our final observation, namely:

1. Two out of the three counterions should be protonated: E136, D140 and E262 have a pKa of 13.37, 9.45 and 5.85, respectively.
2. The remaining aspartates and glutamates are deprotonated: For D130, E141 and D219—pKa of 7.87, 8.18 and 7.85, respectively—the PROPKA results are inconclusive. Even though PROPKA tends more towards protonation, the results are close to 7.0 and especially when considered the PROPKA root-mean-square deviation of 0.89 for glutamates and aspartates (Olssen et al., *J. Chem. Theory Comput.* 2011, 7, 2, 525–537 [DOI: 10.1021/ct100578z]), our choice of leaving these residues deprotonated is within the acceptable margins. Moreover, both D130 and D219 are far away from the active-site region—11 Å and 21 Å, respectively, while in our model E141 is within H-bonding distance of a water molecule and S166, which decreases the likelihood for protonation.